# HER2-driven mammary tumorigenesis enhances bioenergetics despite reductions in mitochondrial content

Sara M Frangos[1]*, Henver S Brunetta[1], Dongdong Wang[2,3], Maria Joy Therese Jabile[2,3], Leslie M Jeffries[4], Grace Mencfeld[4], David WL Ma[1], William J Muller[5], Cezar M Khursigara[6], Kelsey H Fisher-Wellman[7], Jim Petrik[4], Gregory R Steinberg[2,3], Graham P Holloway[1]*

[1]Department of Human Health Sciences, University of Guelph, Guelph, Canada; [2]Centre for Metabolism, Obesity and Diabetes Research, McMaster University, Hamilton, Canada; [3]Division of Endocrinology and Metabolism, Department of Medicine, McMaster University, Hamilton, Canada; [4]Department of Biomedical Sciences, University of Guelph, Guelph, Canada; [5]Department of Biochemistry, McGill University, Rosalind and Morris Goodman Cancer Research, Montreal, Canada; [6]Department of Molecular and Cellular Biology, University of Guelph, Guelph, Canada; [7]Department of Cancer Biology, Medical Center Boulevard, Wake Forest University School of Medicine, Winston-Salem, United States

*For correspondence:
sfrangos@uoguelph.ca (SMF);
ghollowa@uoguelph.ca (GPH)

## eLife Assessment

This **valuable** study aims to determine mechanisms underlying breast cancer initiation and tumour progression. The manuscript includes a **solid** set of transcriptomic and proteomic datasets from tumour samples and examines mitochondrial function within the tumours. While the underlying mechanisms linking expression changes to functional effects remain speculative. This paper provides a resource for researchers working on breast cancer and/or HER2-driven bioenergetics changes.

**Abstract** It is now recognized that mitochondria play a crucial role in tumorigenesis; however, it has become clear that tumor metabolism varies significantly between cancer types. The failure of recent clinical trials aimed at directly targeting tumor respiration through oxidative phosphorylation inhibitors underscores the critical need for further studies providing an in-depth evaluation of mitochondrial bioenergetics. Accordingly, we comprehensively assessed the bulk tumor and mitochondrial metabolic phenotype in murine HER2-driven mammary cancer tumors and benign mammary tissue. Transcriptomic and proteomic profiling revealed a broad downregulation of mitochondrial genes/proteins in tumors, including OXPHOS subunits comprising Complexes I–IV. Despite reductions in tumor mitochondrial proteins, mitochondrial respiration was several-fold higher compared to benign mammary tissue, which persisted regardless of normalization method (wet weight, total protein content, and when corrected for mitochondrial content). This upregulated respiratory capacity could not be explained by OXPHOS uncoupling, suggesting HER2 signaling regulates intrinsic mitochondrial bioenergetics. In further support, lapatinib, an EGFR/HER2 tyrosine kinase inhibitor, attenuated mitochondrial respiration in NF639 murine mammary tumor epithelial cells. Together, this data highlights that the typical correlation between mitochondrial content and respiratory capacity may not apply to all tumor types and implicates HER2-linked activation of mitochondrial respiration supporting tumorigenesis in this model.

## Introduction

The field of cancer metabolism has rapidly evolved in the last decade. A specific focus of many researchers has been on understanding the role of mitochondria in tumorigenesis with the goal to potentially target aspects of mitochondrial metabolism to reduce proliferation (*Vasan et al., 2020*; *DeBerardinis and Chandel, 2020*). It was initially hypothesized that a universal characteristic of cancer cells was dysfunctional oxidative phosphorylation (OXPHOS), as proposed by Otto Warburg in the 1920s (*Warburg, 1924*; *Warburg, 1956*). This theory spurred interest in studying mitochondria in various cancers, highlighting the diverse mitochondrial phenotypes in this context. While instrumental for OXPHOS, mitochondria are now recognized for their multifaceted roles in regulating cellular homeostasis, including supporting macromolecule biosynthesis (*Sullivan et al., 2015*), maintaining redox balance (*Martínez-Reyes et al., 2020*), and generating metabolites that regulate gene transcription (*Morris et al., 2019*). Given their pleiotropic effects within the cell, there is growing recognition that mitochondria play an essential role in cancer initiation, progression, and metastasis (*Vasan et al., 2020*), and an appreciation that interrogating mitochondrial bioenergetics in diverse cancers may identify novel therapeutic strategies.

Reductions in the content/activity of mitochondrial enzymes within the tricarboxylic acid (TCA) cycle and electron transport chain (ETC) often correlate with impaired mitochondrial respiratory capacity and coincide with activating transcriptional programs driving tumorigenesis (*Sciacovelli et al., 2013*; *Cardaci et al., 2015*; *Smith et al., 2020*). Indeed, inhibition of succinate dehydrogenase (Complex II) in osteosarcoma cells (*Sciacovelli et al., 2013*) and renal cell carcinoma (*Cardaci et al., 2015*) enhances cancer growth despite OXPHOS inhibition. Furthermore, in colorectal cancer, age-related Complex I deficiency coincides with reduced respiratory capacity but enhanced serine synthesis and subsequent growth rates (*Smith et al., 2020*). Despite this evidence suggesting that disruptions in mitochondrial bioenergetics contribute to tumor progression, several elegant studies using genetic and pharmacological approaches have solidified that a functional ETC is required for cancer cell growth (*Martínez-Reyes et al., 2020*; *Tan et al., 2015*; *Weinberg et al., 2010*). Furthermore, the use of whole-genome sequencing specifically interrogating mtDNA phenotypes across several human cancers has revealed that loss of function mutations in mtDNA genes are often under negative selection (*Yuan et al., 2020*), suggesting the maintenance of mitochondrial function may support tumorigenesis in some cancers and impairments in mitochondrial metabolism are not a ubiquitous finding. However, mitochondrial content represents one dimension of bioenergetic regulation and does not alone determine respiratory capacity. Therefore, a comprehensive assessment of mitochondrial function, particularly within the context of the mitochondrial proteome, is warranted to provide deeper insights into the relationship between mitochondrial content and bioenergetics in different cancers.

Within this framework, there has been considerable interest in investigating mitochondrial bioenergetics and dynamics in breast cancer subtypes due to the limited therapeutic options for triple-negative breast cancer (TNBC) and therapeutic resistance in breast cancers driven by estrogen (ER), progesterone (PR), and human epidermal growth factor (HER2) receptor amplification (*Wang et al., 2022*; *Avagliano et al., 2019*; *Baek et al., 2023*). Indeed, increases in mitochondrial respiration or 'OXPHOS reliance' are associated with poor outcomes in treatment-naive TNBC (*Evans et al., 2021*), metastatic ER+ tumors (*El-Botty et al., 2023*), and HER2-driven breast cancers (*Rohlenova et al., 2017*). While inhibiting OXPHOS in this context (*Evans et al., 2021*; *El-Botty et al., 2023*) (i.e. with Complex I inhibitor IACS-010759) has shown some preclinical success, systemic toxicity has prevented the translation of this strategy to the clinic (*Yap et al., 2023*). Furthermore, it is unclear whether increases in mitochondrial respiration arise from increases in mitochondrial content or intrinsic enhancements in mitochondrial function (e.g. coupling efficiency, substrate delivery/sensitivity, oncogenic signaling events). This information is critical to develop successful mitochondrial-targeted therapies with a wider therapeutic window. Thus, in the current study, we comprehensively assessed the bulk tumor and mitochondrial phenotype between murine HER2-driven mammary cancer tumors and benign mammary tissue in the context of the underlying transcriptome and proteome. Respiratory capacity supported by both pyruvate and lipids was enhanced in HER2-driven tumors despite lower mitochondrial content compared to benign mammary tissue. Mitochondrial bioenergetics were suppressed by the dual tyrosine kinase inhibitor lapatinib, suggesting events linked to HER2-mediated oncogenic signaling may drive increases in OXPHOS during tumorigenesis.

## Results

### Mammary tumors from MMTV/neu[ndl]-YD5 mice display canonical HER2 pathway activation

Cell lines are commonly used to study isolated mechanisms driving tumorigenesis; however, they neglect the variety of cell types in the tissue of origin and the influence of the tumor microenvironment, which is a key regulator of cancer growth (*Mun et al., 2024*; *de Visser and Joyce, 2023*). To this end, we have utilized tumor and benign mammary tissues from MMTV/neu[ndl]-YD5 transgenic mice (*Ursini-Siegel et al., 2007*) to investigate HER2-driven tumor biology (*Figure 1A*). Signaling events linked to HER2 activation rely on the autophosphorylation of the tyrosine kinase domain (*Ursini-Siegel et al., 2007*; *Iqbal and Iqbal, 2014*; *Dankort et al., 1997*), and MMTV/neu[ndl]-YD5 mice possess an add-back mutation at the Y1226/7 (YD) site, activating downstream signaling events linked to SHC/GRB2 with similar transforming potential as the wild-type protein (*Dankort et al., 1997*; *Dankort et al., 2001*; *Figure 1B*). While the proximal signaling events linked to HER2 activation in this model have been described (*Dankort et al., 1997*), to validate our bulk-tissue-based model, we assessed the mRNA (transcriptomics) and protein (quantitative label-free proteomics and western blotting) expression of canonical HER2 pathway members (RAS/MAPK and PI3K/AKT/mTOR pathways) that are well-characterized in various HER2-overexpressing mammary cancer cell lines (*Figure 1B*; *Ruiz-Saenz et al., 2018*; *Kirouac et al., 2016*; *Serra et al., 2011*). *Her2* transcript levels were increased ~175-fold (*Figure 1C*) and transcripts belonging to the Ras/MAPK (*Hras*, *Mapk1*, *Mapk3*) and PI3K/AKT/mTOR (*Pik3cg*, *Pik3c2a*, *Pik3c2b*, *Pik3c2g*, *Pik3c3*, *Akt3*) pathways were also upregulated in the tumors compared to the benign mammary tissue (*Figure 1D*). At the protein level, HER2 was readily detectable in mammary tumors by quantitative label-free proteomics but undetectable in paired benign tissues (*Figure 1E*), suggesting enhanced HER2 transcription also manifests as an increase in receptor content in the tumors. Downstream of HER2 activation, the total protein content of p38 MAPK, ERK1/2, and eEF2 were significantly increased in tumors compared to benign tissue detected by western blot (*Figure 1F, G*). When normalized to total protein, p38 MAPK and eEF2 phosphorylation were not higher in tumors due to proportional increases in total protein abundance, while in contrast, phosphorylated ERK1/2 and mTOR were significantly upregulated, suggesting both Ras/MAPK and the PI3K/AKT/mTOR pathways were activated in tumors (*Figure 1H*, *Figure 1—figure supplement 1*). Overall, these data demonstrate that mammary tumors display classical hallmarks of HER2-driven cancer cell signaling, allowing for the subsequent investigation of HER2-driven tumor biology and metabolism in this model.

### Assessment of the HER2-driven mammary tumor transcriptome

To understand the biology and metabolism of HER2-driven mammary tumors, we assessed their transcriptomes using RNA sequencing. Principal component analysis (PCA) using the entire gene expression profile showed distinct separation between groups (*Figure 2A*). Visualizing all differentially expressed genes (DEGs) using a volcano plot (*Figure 2B*) and the top 3000 DEGs using a heatmap (*Figure 2C*) revealed several differentially upregulated and downregulated genes between groups where genes involved in MAPK- and AKT-linked signaling, ECM remodeling, and cellular stress responses (*Etv1*, *Dusp6*, *Papln*, *Ier3*) were among the most upregulated genes in the tumors and genes involved in lipid metabolism (*Dgat2*, *Lipe*, *Cd36*) were among the most downregulated. Enrichment analysis of the top 3000 upregulated DEGs (*Figure 2D*) identified pathways characteristic of HER2-driven signaling (e.g. tyrosine kinase signaling, GTPase signaling) and gene ontology (GO) analysis identified processes indicative of active transcription, translation, and protein turnover characteristic of a rapidly proliferating tissue (*Figure 2E*). Interestingly, enrichment analysis of the top 3000 downregulated genes in the tumors consistently identified that mitochondrial metabolism, and in particular components of the oxidative phosphorylation (OXPHOS) system, were downregulated in tumors compared to benign mammary tissue (*Figure 2F, G*).

### Assessment of HER2-driven mammary tumor proteome

We further validated our bulk-tissue transcriptomic findings at the protein level using label-free quantitative proteomics and detected 2589 proteins in the benign mammary tissue and 2448 proteins in the HER2-driven tumors; 1848 of which were found in both groups (*Figure 3A*). This initial

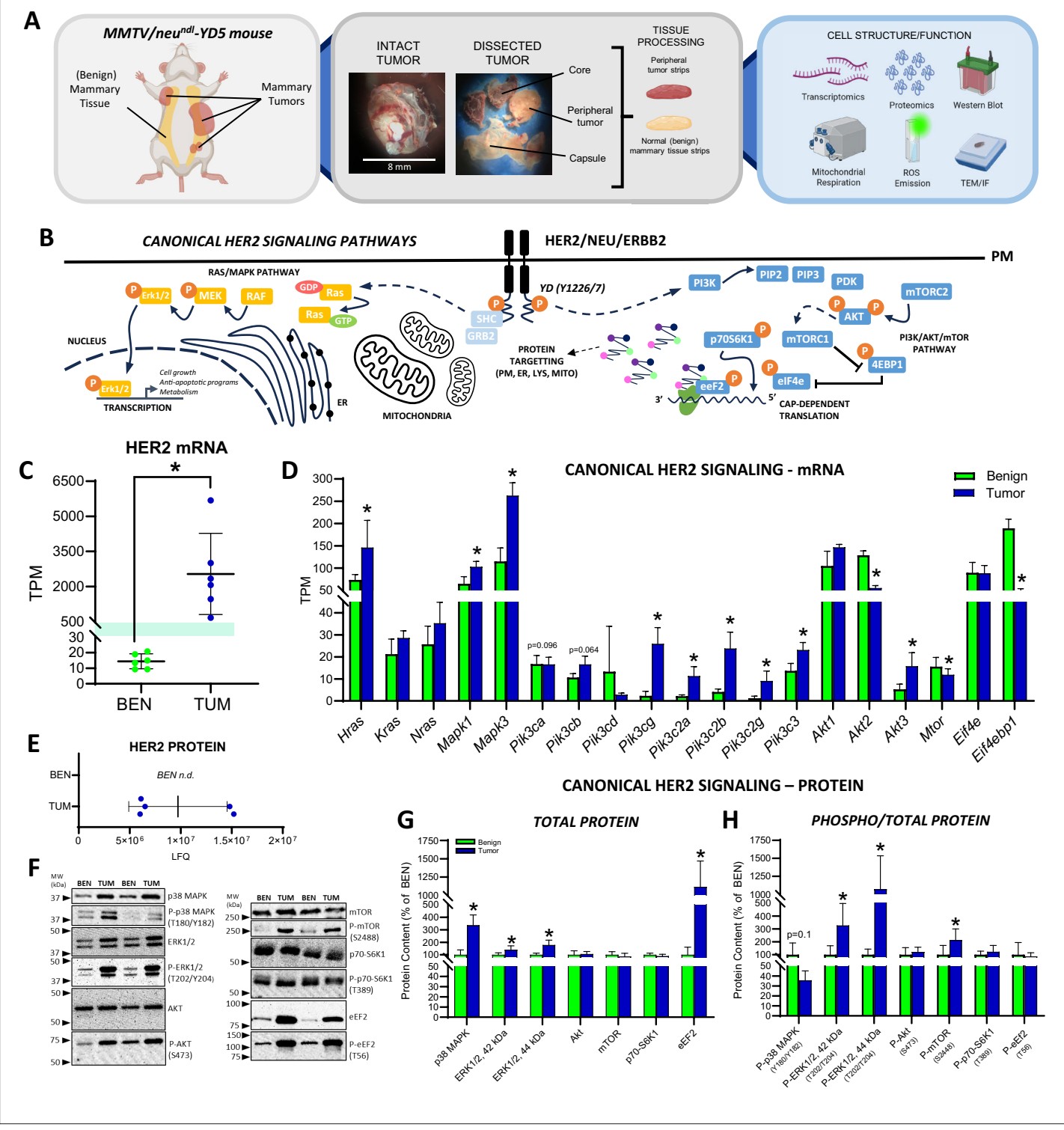

**Figure 1.** Experimental design and canonical HER2 signaling. (**A**) Schematic of mammary tumors from the mammary fat pad of female MMTV-neu[ndl]-YD5 mice and normal (benign) mammary tissue, representative images of tumor morphology (tumor capsule, peripheral tumor tissue, and necrotic core), tissue processing workflow for transcriptomics, proteomics, western blotting, mitochondrial respiration, ROS emission, and histology (transmission electron microscopy (TEM) and immunofluorescence (IF)). (**B**) Schematic of classical signaling pathways (Ras/MAPK and PI3K/AKT/mTOR) downstream of HER2 receptor stimulation in the YD5 model. (**C**) *Her2* gene expression (transcripts per million, TPM). (**D**) Gene expression for canonical HER2 signaling. (**E**) Absolute abundance of HER2 protein detected by quantitative label-free proteomics (n.d. = HER2 not detected in benign). (**F**) Representative western blots of proteins involved in canonical HER2 signaling. Quantified western blots (% of benign) for (**G**) total protein

*Figure 1 continued on next page*

*Figure 1 continued*

content and (**H**) phosphorylated targets normalized to the total protein content of each target. p-values in panels **D** and **E** listed as adjusted p-values from transcriptomic (differentially expressed gene, DEG) data (p-adj. <0.1 considered statistically significant). In panels **G, H** data is presented as means ± SD and statistical significance was determined using an unpaired, two-tailed Student's *t*-test (*p < 0.05, *n* = 5–6 biological replicates per group/protein target). *Abbrev. AKT – protein kinase B; eEf2 – eukaryotic elongation factor 2; eIF4E – eukaryotic translation initiation factor 4E; ERK – extracellular signal-regulated kinase; GRB2 – growth factor receptor-bound protein-2; MEK – mitogen-activated protein kinase kinase; mTORC1/2 – mechanistic target of rapamycin complex 1 and 2; PDK – 3-phosphoinositide-dependent protein kinase-1; PI3K – phosphoinositide 3-kinase; PIP2/3 – phosphatidylinositol 4,5-bisphosphate; PIP3 – phosphatidylinositol 3,4,5-triphosphate; p70-S6K1 – ribosomal protein S6 kinase beta-1; RAF – rapidly accelerated fibrosarcoma; RAS – rate sarcoma virus; SHC – Src homolog and collagen homolog; 4EBP1 – eukaryotic translation initiation factor 4E-binding protein 1.*

The online version of this article includes the following source data and figure supplement(s) for figure 1:

**Source data 1.** PDF file containing original western blots for *Figure 1F* with annotated bands and molecular weight markers.

**Source data 2.** Original TIF files for western blot images in *Figure 1F–H*.

**Figure supplement 1.** Absolute phosphorylated protein targets downstream of HER2 activation.

**Figure supplement 1—source data 1.** PDF file containing original western blots for *Figure 1—figure supplement 1A, B* with annotated bands and molecular weight markers.

**Figure supplement 1—source data 2.** Original TIF files for western blot images in *Figure 1—figure supplement 1A, B*.

analysis revealed 741 and 600 uniquely expressed (i.e. only detected in one tissue) proteins in benign mammary tissue and tumors, respectively (*Figure 3—figure supplement 1A*). Interestingly, OXPHOS and aerobic respiration were identified among the top pathways in benign tissue, but not in tumors (*Figure 3—figure supplement 1B–G*), further supporting the notion of downregulated mitochondrial content within tumors. Reflecting the PCA for the transcriptomic data, analysis of all overlapping proteins between benign and tumor samples revealed distinct separation (*Figure 3B*). Of the 1848 proteins found in both the tumors and benign tissue, 420 were significantly different (p-adj. <0.1, *Figure 3A, C, D*). However, the majority of the differentially expressed proteins were downregulated (*n* = 411) in tumors compared to benign mammary tissue. Upregulated proteins (*n* = 9) included those involved in canonical HER2 signaling, protein synthesis and processing, and glycolytic metabolism (*Supplementary file 1*). Supporting transcriptomic data, which indicated a reduction in mitochondrial content and OXPHOS, enrichment analysis revealed that 8 of the top 10 downregulated pathways (*Figure 3E*) and all top 10 downregulated GO (*Figure 3F*) were related to mitochondrial biogenesis and/or metabolism, and OXPHOS was identified in the top 10 downregulated Kyoto Encyclopedia of Genes and Genomes (KEGG) pathways (*Figure 3G*). Together, these proteomic data support our transcriptomic analysis suggesting HER2-driven tumors are characterized by lower mitochondrial content than benign mammary tissue.

## Integrated assessment of the HER2-driven tumor transcriptome and proteome

The concordant nature of our transcriptomic and proteomic datasets was highlighted when correlating the log$_2$FC between overlapping gene and protein targets (*Figure 4A*). Sorting these data by mitochondrial proteins highlighted that the majority of overlapping mitochondrial proteins detected in both datasets were downregulated (*Figure 4B*). Thus, we further analyzed the 144 overlapping downregulated targets between these analyses (*Figure 4C*). KEGG enrichment analysis of the downregulated targets identified OXPHOS among the top 10 pathways, as well as carbon metabolism, BCAA degradation, and fatty acid metabolism, which encompass numerous mitochondrial targets (*Figure 4D*). The top 10 common downregulated pathways (*Figure 4E*) and GO (*Figure 4F*) all pertained to mitochondrial substrate metabolism/catabolism and OXPHOS. Together, transcriptomic and proteomic analyses strongly support that HER2-driven tumors are characterized by downregulated mitochondrial content.

## Assessment of the mitochondrial transcriptome and proteome

Our initial transcriptomic and proteomic analyses encompassed all DEGs/proteins between tissues (including mitochondrial targets) and clearly identified reductions in mitochondrial content in the tumors. This emphasized the robust nature of mitochondrial adaptations in this model, considering

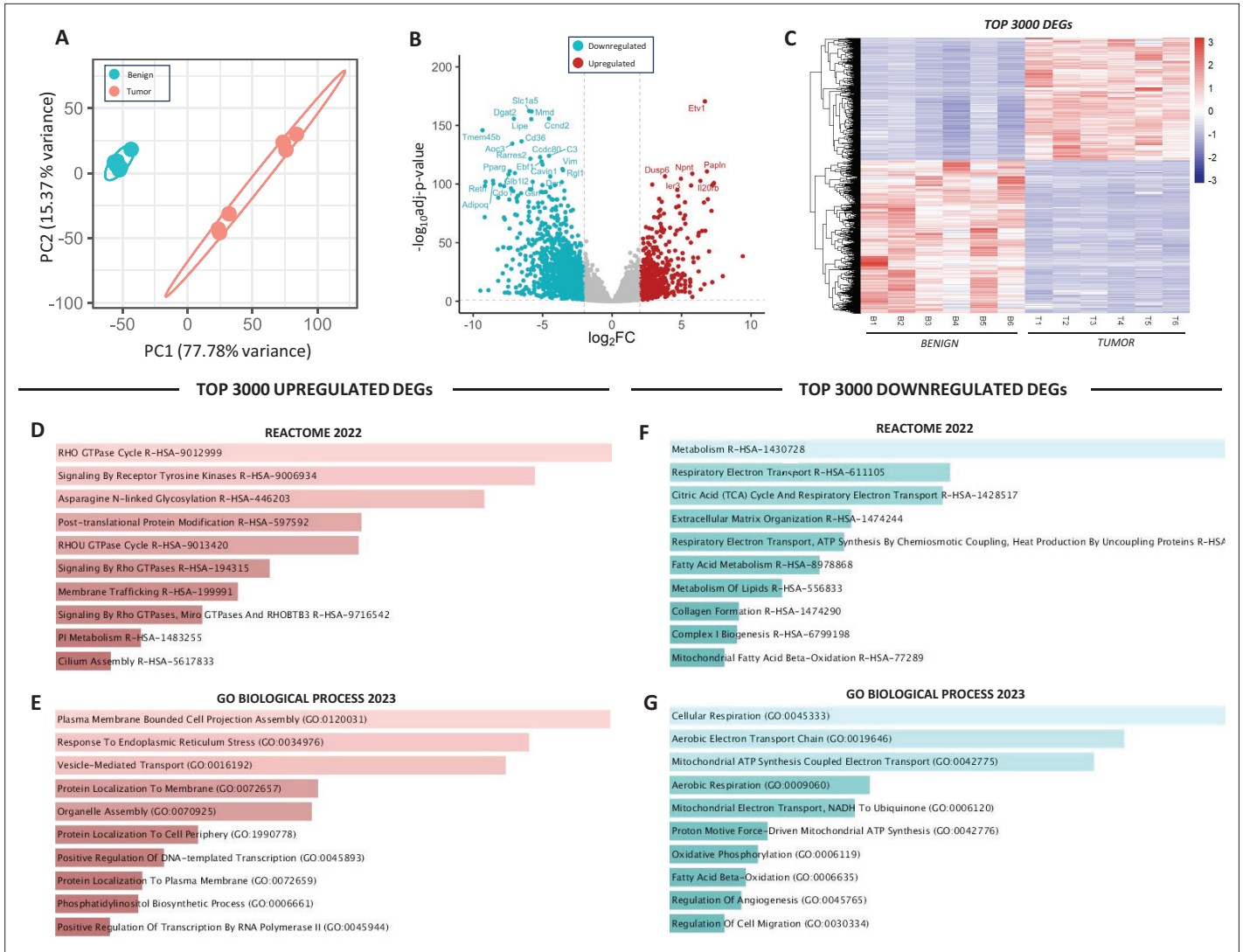

**Figure 2.** Assessment of HER2-driven mammary tumor and benign mammary tissue transcriptomes. (**A**) Principal component analysis (PCA) of benign mammary tissue and mammary tumors (*n* = 6 biological replicates/group). (**B**) Volcano plot showing differentially expressed genes (DEGs) identified between mammary tissue and mammary tumors. (**C**) Heatmap of top 3000 DEGs, (**D**) top 10 upregulated pathways (Reactome 2022) and (**E**) gene ontologies (GO Biological Process 2023) in tumors compared to benign mammary tissue, (**F**) top 10 downregulated pathways (Reactome 2022) and (**G**) gene ontologies (GO Biological Process 2023) in tumors compared to benign mammary tissue. Panels **D, E** assess the top 3000 upregulated DEGs (log₂fc >1), panels **F, G** assess the top 3000 downregulated DEGs (log₂fc <1). An adjusted p-value of <0.1 was considered statistically significant across all analyses. The Enrichr open access analysis web tool was used to generate panels **D–G** which were sorted by p-value.

mitochondrial proteins (identified using the mouse MitoCarta 3.0 database) represented a small fraction of total cellular protein in both tumors and benign mammary tissue (***Figure 5—figure supplement 1A***). By filtering our proteomic dataset using MitoCarta, we identified 125 mitochondrial proteins that were commonly expressed between the tumors and benign mammary tissue (***Figure 5A***). When visualizing the 125 proteins with a heatmap, the majority of the mitochondrial proteome was downregulated in the tumors compared to the benign mammary tissue (***Figure 5B***). The summed absolute abundance of all detected mitochondrial proteins in each group was also lower in tumors (***Figure 5C***), again supporting our transcriptomic and proteomic analyses suggesting a reduction in mitochondrial content exists in the tumors.

While mitochondrial biogenesis is a coordinated process simultaneously upregulating several mitochondrial proteins, the total mitochondrial transcriptome/proteome does not necessarily change proportionally with OXPHOS complex and/or TCA cycle/dehydrogenase content, especially in cancerous tissues. Indeed, reductions in the content of certain respiratory complexes seem to be

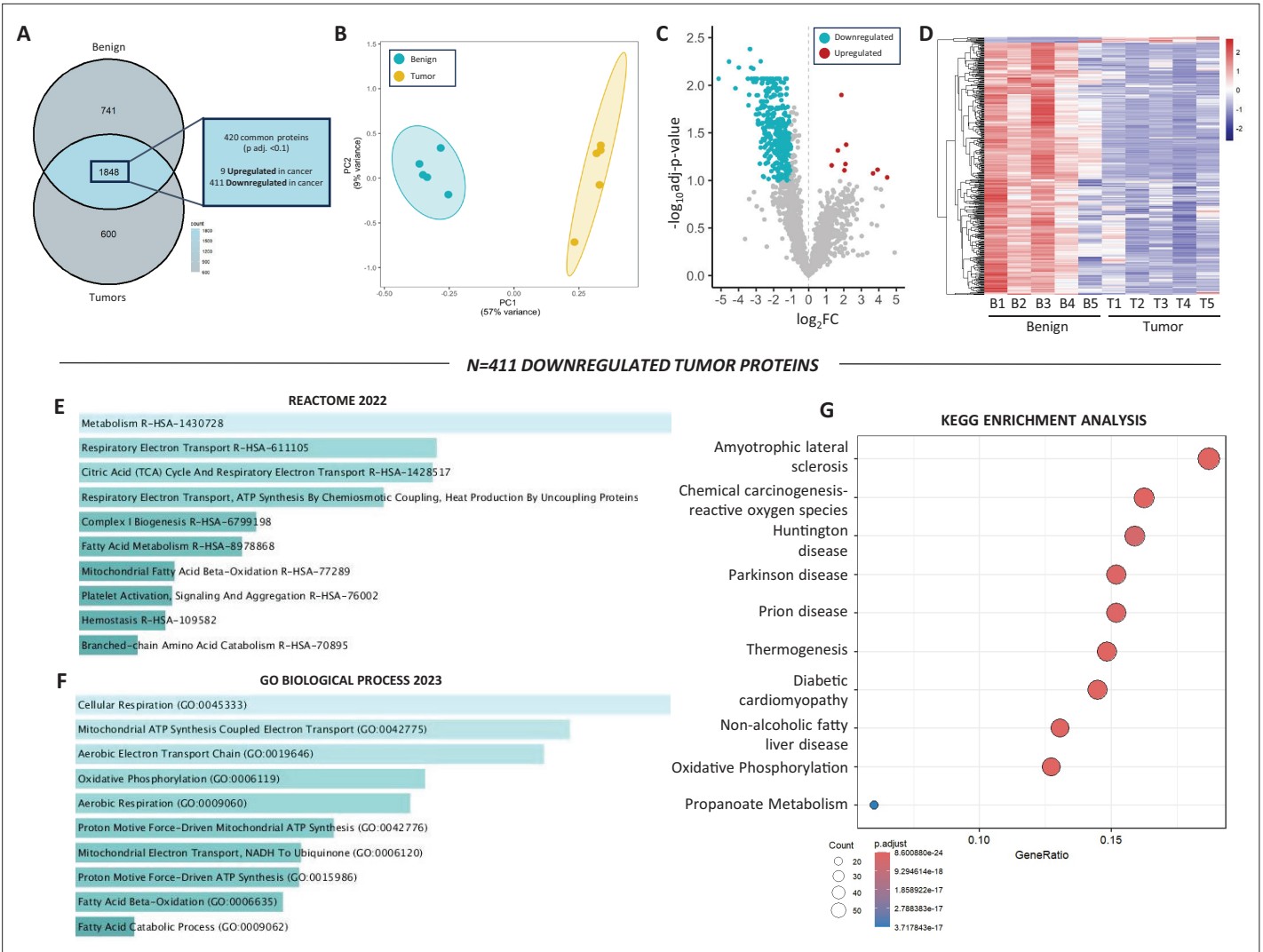

**Figure 3.** Assessment of HER2-driven mammary tumor and benign mammary tissue proteomes. (**A**) Venn diagram showing all detected proteins. Of the 1848 overlapping targets, 420 were significantly different based on a p-adj. <0.1 and 411 of those proteins were downregulated in tumors. (**B**) Principal component analysis (PCA) of benign mammary samples and mammary tumors. (**C**) Volcano plot showing proteins identified between mammary tumors and benign mammary tissue, those in blue (downregulated) or red (upregulated) were significantly different ($n$ = 420 proteins, p-adj. <0.1). (**D**) Heatmap of 420 differentially expressed proteins in benign and tumor samples. (**E**) Top 10 downregulated pathways (Reactome 2022) and (**F**) gene ontologies (GO Biological Process, 2023), (**G**) Kyoto Encyclopedia of Genes and Genomes (KEGG) analysis of downregulated proteins. The Enrichr open access analysis web tool was used to generate panels **E, F**, which were sorted by p-value.

The online version of this article includes the following figure supplement(s) for figure 3:

**Figure supplement 1.** Unique HER2-driven tumor and benign mammary tissue proteins.

cancer specific, which can have functional repercussions on mitochondrial respiration rates and/or efficiency (*Smith et al., 2020*; *Boykov et al., 2023*). Within this context, we assessed the gene expression profile of all OXPHOS complex subunits between groups, which revealed relatively ubiquitous reductions in transcript count (transcripts per million) across subunits of Complexes I–IV in the tumors (*Figure 5D–H*). Interestingly, while the majority of Complex V subunits were also downregulated, this complex had the highest proportion of subunits displaying no differences between groups (*Figure 5D, I*). The TCA cycle transcriptome also reflected a general downregulation of transcript abundance, apart from alpha-ketoglutarate dehydrogenase (*Ogdhl*) and succinyl-CoA synthetase GDP forming subunit-β (*Suclg2*), which had a higher transcript abundance in some tumor samples (*Figure 5J*). While we focused on transcriptomic data for this analysis due to lower detection of certain OXPHOS subunits using label-free quantitative proteomics (*Figure 5—figure supplement*

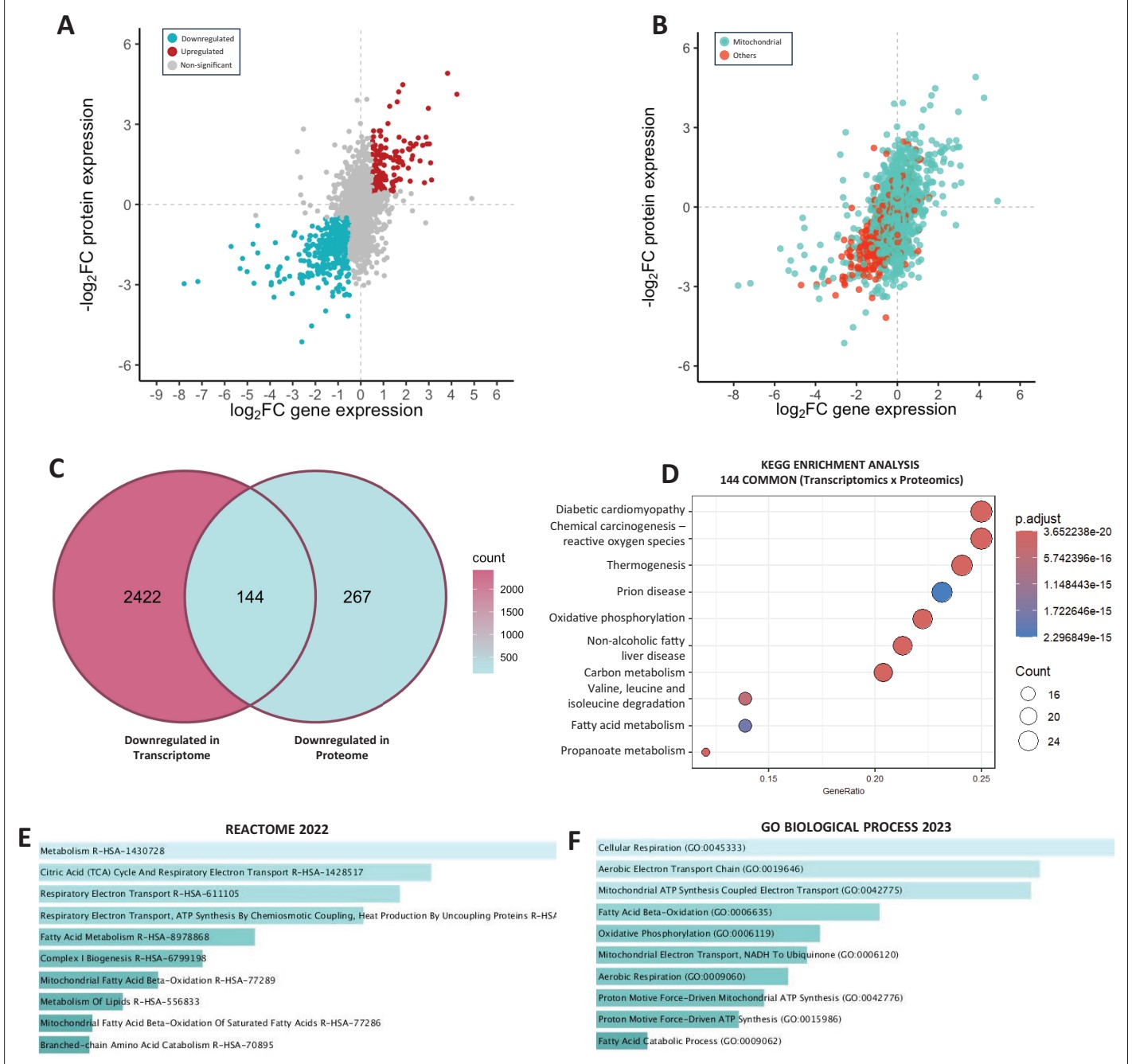

**Figure 4.** Overlapping genes and proteins between HER2-driven mammary tumor transcriptome and proteome. Correlation of log$_2$FC for overlapping targets between the HER2-driven tumor and benign mammary tissue transcriptomes and proteomes grouped by up/downregulation (**A**), and (**B**) protein localization (mitochondrial vs other). (**C**) Venn diagram showing 144 overlapping downregulated targets between the assessed HER2-driven tumor and benign mammary tissue transcriptomes and proteomes. (**D**) Kyoto Encyclopedia of Genes and Genomes (KEGG) analysis for common downregulated targets. Top 10 downregulated (**E**) pathways (Reactome 2022) and (**F**) gene ontologies (GO Biological Process 2023). An adjusted p-value of <0.1 was utilized for all analyses. The Enrichr open access analysis web tool was used to generate panels **E, F**, which were sorted by p-value.

*2A–E*), the summed abundance of all OXPHOS subunits detected per complex expressed relative to the underlying mitochondrial proteome (*Figure 5—figure supplement 2F*), and relative to the total proteome (*Figure 5—figure supplement 2G*) displayed a similar pattern. This interpretation was further validated by western blot, where tumor CII and CIV content were lower than benign mammary tissue (*Figure 5—figure supplement 2H*). Overall, these data further suggest that the mitochondrial proteome and OXPHOS Complexes I–IV are downregulated in HER2-driven tumors.

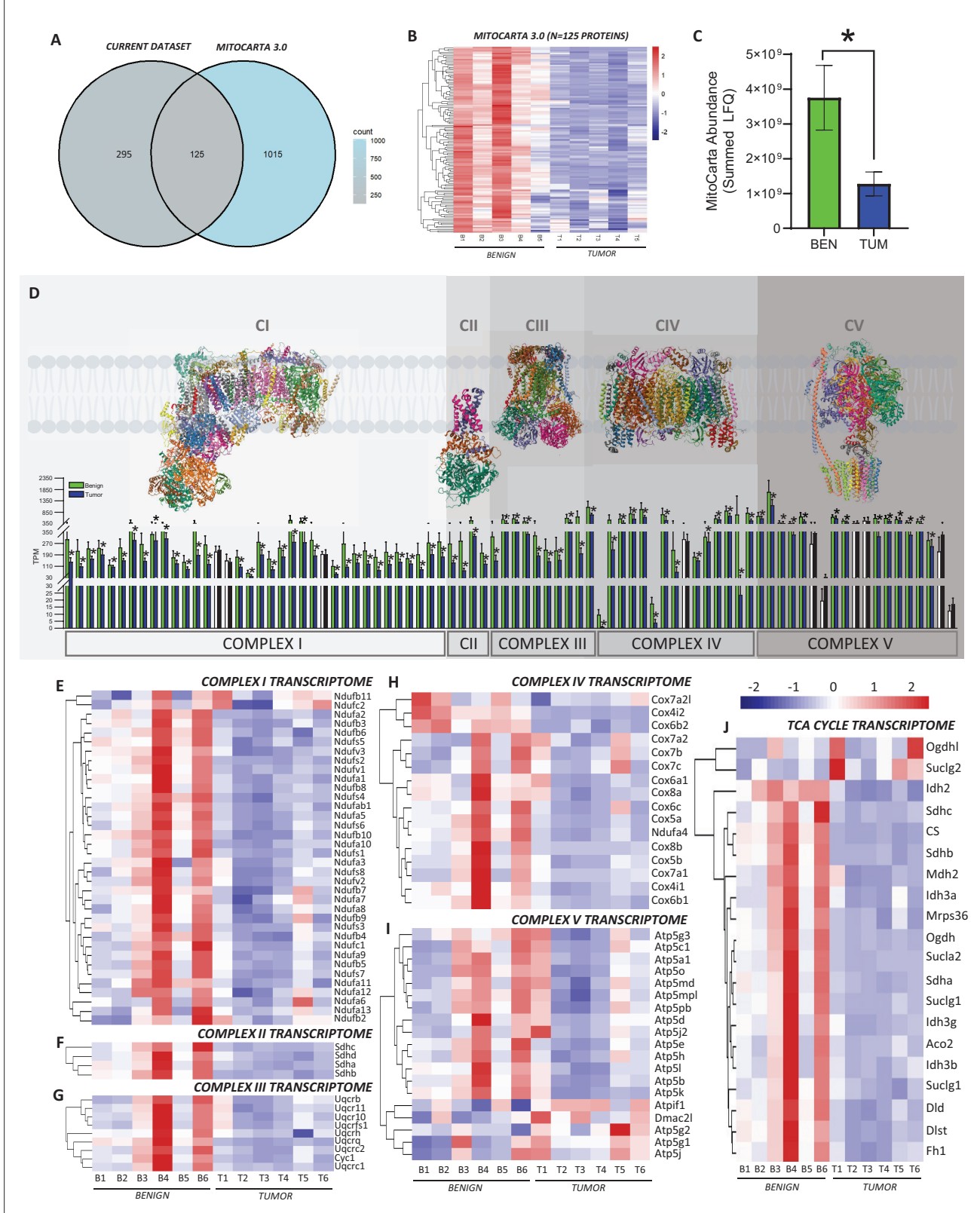

**Figure 5.** Assessment of the mitochondrial transcriptome and proteome. (**A**) Venn diagram showing mitochondrial proteins detected in the current dataset when cross-referenced with the Mouse MitoCarta 3.0 database. (**B**) Heatmap of 125 mitochondrial proteins identified in benign mammary tissue and HER2-driven tumors (*n* = 5 biological replicates/group). (**C**) Absolute abundance of all mitochondrial proteins detected in benign and tumor tissues. (**D**) Gene expression (detected by transcriptomics; transcripts per million) of all detected OXPHOS subunits separated per respiratory complex;

*Figure 5 continued on next page*

Figure 5 continued

black and white bars represent subunits with no significant differences between groups. Heatmaps of (**E**) Complex I, (**F**) Complex II, (**G**) Complex III, (**H**) Complex IV, (**I**) Complex V (ATP synthase), and (**J**) tricarboxylic acid (TCA) cycle transcriptomes. All proteomic analyses (panels A–C) were conducted on $n = 5$ biological replicates/group. All transcriptomic analyses (panels D–J) were conducted on $n = 6$ biological replicates/group. Panels B, C were analyzed using an unpaired two-tailed Student's *t*-test (*p < 0.05). In panel D, * and colored (blue and green bars) represent significant differences between benign and tumor samples based on an adjusted p-value of <0.1. OXPHOS complex schematics in panel D were obtained from the open access RCSB Protein Data Bank (*Complex I = TW1V, Complex II = 2A06, Complex III = 3SFD, Complex IV = 6NMP, Complex V = 5FIL*). Bars in panel **D** *are listed in the following order: Complex I: Ndufa1, Ndufa10, Ndufa11, Ndufa12, Ndufa13, Ndufa2, Ndufa3, Ndufa5, Ndufa6, Ndufa7, Ndufa8, Ndufa9, Ndufab1, Ndufb10, Ndufb11, Ndufb2, Ndufb3, Ndufb4, Ndufb5, Ndufb6, Ndufb7, Ndufb8, Ndufb9, Ndufc1, Ndufc2, Ndufs1, Ndufs2, Ndufs3, Ndufs4, Ndufs5, Ndufs6, Ndufs7, Ndufs8, Ndufv1, Ndufv2, Ndufv3; Complex II: Sdha, Sdhb, Sdhc, Sdhd; Complex III: Cyc1, Uqcr10, Uqcr11, Uqcrb, Uqcrc1, Uqcrc2, Uqcrfs1, Uqcrh, Uqcrq; Complex IV: Cox4i1, Cox4i2, Cox5a, Cox5b, Cox6a1, Cox6b1, Cox6b2, Cox6c, Cox7a1, Cox7a2, Cox7a2l, Cox7b, Cox7c, Cox8a, Cox8b, Ndufa4; Complex V: Atp5a1, Atp5b, Atp5c1, Atp5d, Atp5e, Atp5g1, Atp5g2, Atp5g3, Atp5h, Atp5j, Atp5j2, Atp5k, Atp5l, Atp5md, Atp5mpl, Atp5o, Atp5pb, Atpif1, Dmac2l.*

The online version of this article includes the following source data and figure supplement(s) for figure 5:

**Figure supplement 1.** Mitochondrial enrichment in benign mammary tissue and HER2-driven mammary tumors.

**Figure supplement 2.** Proteomic analysis of detected OXPHOS subunits.

**Figure supplement 2—source data 1.** PDF file containing original western blots for *Figure 5—figure supplement 2H* with annotated bands and molecular weight markers.

**Figure supplement 2—source data 2.** Original TIF files of western blot images in *Figure 5—figure supplement 2H*.

---

While these findings strongly support reduced mitochondrial gene expression in tumors, differences could theoretically result from shifts in cell-type during tumorigenesis. To address this, we first assessed OXPHOS expression in a publicly available single-cell RNA sequencing (scRNA-seq) dataset from human breast tumors (*Wu et al., 2021*). In the scRNA-seq dataset, tumor epithelial cells expressed the highest proportion of CI-CIV subunits when grouped by cell type (*Figure 6A, B*). We next assessed cell-type composition in our tumor and benign mammary tissue samples by immunofluorescence (*Figure 6C*). Although the immunopositive area was lower across all cell types in benign samples, mesenchymal and epithelial cells remained the dominant populations in both groups, indicating the absence of major shifts in the most abundant cell types (*Figure 6D*). To integrate cell-type-specific OXPHOS mRNA expression (scRNA-seq) with cell-type abundance in our samples, we calculated a cell-type-weighted OXPHOS mRNA Index, reflecting each population's estimated contribution to Complex I–V transcript levels in our tissues. The OXPHOS mRNA Index (*Figure 6E*) indicated that epithelial cells were likely the dominant contributors to OXPHOS transcripts in the present tumor samples. When these values were summed across complexes to generate a Total OXPHOS mRNA Score, epithelial cells remained the primary contributors (*Figure 6F*). Altogether, these analyses suggest that our whole-tissue RNA-seq signatures are consistent with canonical HER2-driven signaling and indicate that the reduction in OXPHOS gene expression observed by bulk-tissue RNA-seq likely cannot be explained by a loss of high-OXPHOS epithelial cells or a gain in low-OXPHOS cell populations. Instead, these findings support a cell-intrinsic suppression of mitochondrial gene expression in HER2-driven tumors.

## Mitochondrial respiratory capacity is increased in HER2-driven mammary tumors

Considering our transcriptomic and proteomic data strongly suggest HER2-driven mammary tumors display reductions in mitochondrial content, we next assessed whether this would manifest as a reduction in respiratory capacity by measuring mitochondrial respiratory flux rates. In permeabilized tumors and benign mammary tissue, we assessed basal/endogenous respiration and respiration with saturating carbohydrate (pyruvate/malate) and lipid (palmitoyl-carnitine) substrates. For each protocol, several biological replicates (individual tumors) were utilized, spanning a continuum of whole tumor wet weights/volumes (*Figure 7—figure supplement 1A, B*). Since no strong correlations were found between tumor wet weight or volume and basal or maximal respiration supported by carbohydrate- or lipid-linked substrates (*Figure 7—figure supplement 1C–J*), all data points were pooled. When normalizing respiration to tissue wet weight (*Figure 7A*), tumors displayed significantly higher basal/endogenous respiration and higher respiration with subsequent additions of pyruvate, malate (CI-linked respiration), ADP (State III respiration), glutamate (CI-linked respiration

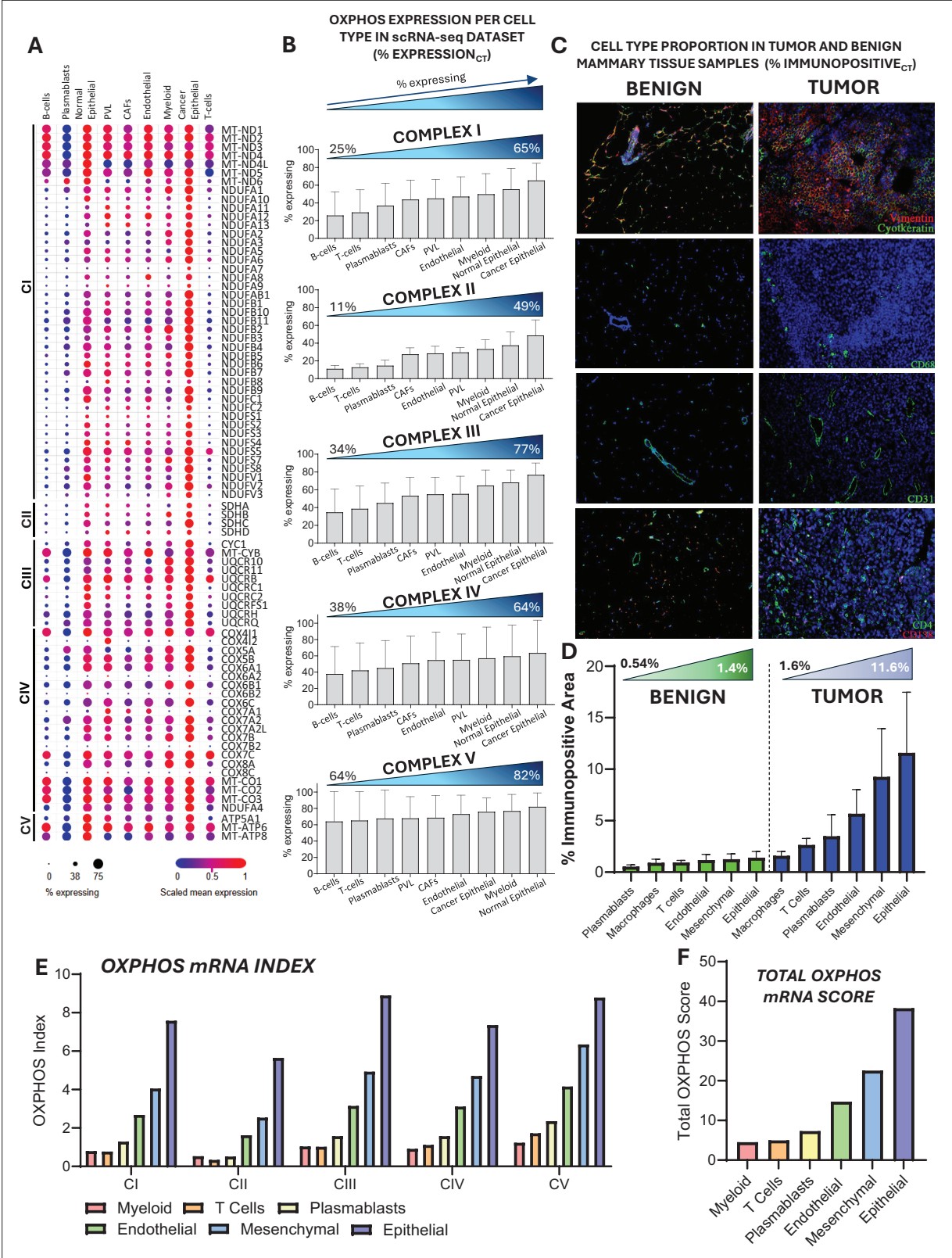

**Figure 6.** Cell-type-specific OXPHOS mRNA analysis in benign and tumor samples. (**A**) OXPHOS expression per cell type from a publicly available single-cell RNA sequencing dataset (**Wu et al., 2021**) shown as a dot plot for each respiratory complex (Complexes I–V) subunit that was detected. Dot size indicates the percentage of cells of a given major type expressing each subunit gene (% cells expressing). Dot color represents gene expression scaled relative to each gene's expression distribution across all major cell types. (**B**) Percentage of cells expressing OXPHOS subunits, arranged in

*Figure 6 continued on next page*

*Figure 6 continued*

ascending order per cell type for Complexes I–V. (**C**) Representative immunofluorescence images of benign mammary tissue and tumor samples stained for vimentin, cytokeratin, CD68, CD31, CD4, and CD138. (**D**) Quantification of cell-type proportion (% immunopositive area) in benign and tumor samples. (**E**) OXPHOS mRNA Index calculated for Complexes I–V. $OXPHOS\ mRNA\ Index = (\%Expression_{CT}) * (\%Immunopositive_{CT})$, where CT represents cell type. (**F**) Total OXPHOS mRNA Score for each cell type. $TOTAL\ OXPHOS\ mRNA\ Score = \sum_{i=I,II,III,IV,V} OXPHOS\ mRNA\ Index\ Ci$, where $i$ represents mitochondrial respiratory complexes.

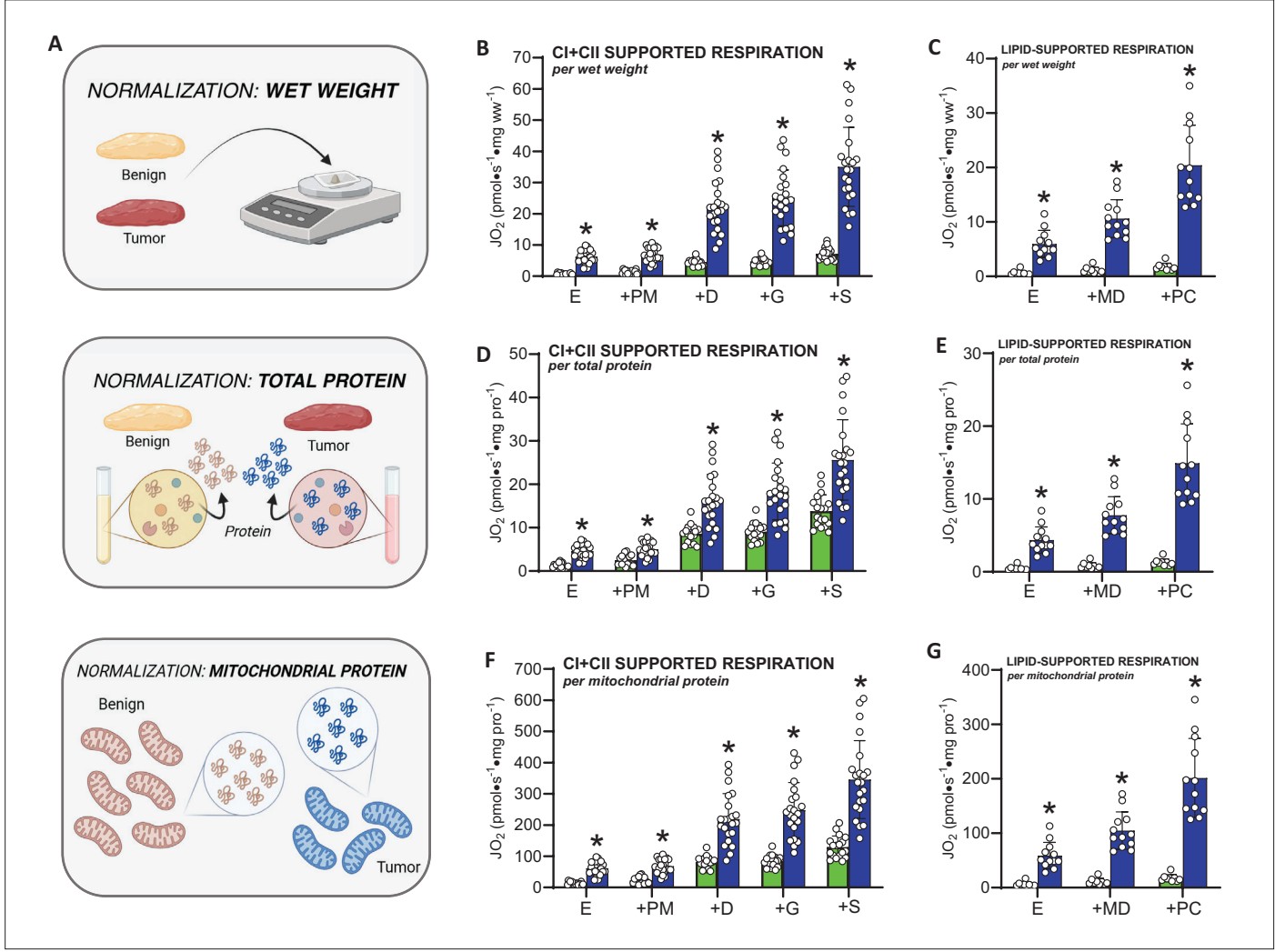

**Figure 7.** Assessment of lipid and carbohydrate-linked mitochondrial respiration and OXPHOS coupling in benign mammary tissue and HER2-driven mammary tumors. (**A**) Schematic depicting methods of normalization for respiration rates (*per wet weight*, *total protein*, *mitochondrial protein/enrichment*). (**B**) Complex I- and II-supported respiration normalized to tissue wet weight, (**C**) lipid-supported respiration normalized to tissue wet weight, (**D**) Complex I- and II-supported respiration normalized to total protein content, (**E**) lipid-supported respiration normalized to total protein content, and (**F**) Complex I- and II-supported respiration normalized to mitochondrial protein (mitochondrial enrichment factor, MEF). (**G**) Lipid-supported respiration normalized to mitochondrial protein. All individual values represent biological replicates (*n* = 9–22 biological replicates/group depending on the protocol). Data are presented as means ± SD, statistical analyses were conducted using an unpaired, two-tailed Student's *t*-test (*p < 0.05). *Abbrev: E = Endogenous, PM = pyruvate (5 mM), malate (2 mM), D = ADP (5 mM), G = glutamate (10 mM), S = succinate (10 mM), PC = palmitoyl-carnitine (20 µM), CI + CII = maximal CI + CII-supported respiration (PMDGS).*

The online version of this article includes the following figure supplement(s) for figure 7:

**Figure supplement 1.** Correlation of tumor size and volume with maximal carbohydrate and lipid-supported respiration rates.

also supported by alpha-ketoglutarate anaplerosis), and succinate (to maximize CII-linked respiration) (*Figure 7B*). Equally, tumors displayed significantly higher respiration rates supported by lipids (palmitoyl-carnitine, *Figure 7C*), suggesting that HER2-driven tumors are capable of relying on both substrate sources, perhaps to maximize aerobic metabolism. This initial analysis of respiratory rates was normalized to tissue wet weight, which can be influenced by differences in total and mitochondrial protein content in a given tissue volume. To address this, we first normalized respiratory flux rates to total cellular protein (*Figure 7D, E*), which minimizes differences in factors such as tissue density and hydration that are not directly contributing to mitochondrial oxygen consumption. The third normalization method employed was normalizing respiratory flux to a mitochondrial enrichment factor (MEF) (*Figure 7F, G*) which is the ratio of mitochondrial proteins to the underlying total proteome (*Figure 5—figure supplement 1B*). This approach controls for differences in mitochondrial content between tissues to determine whether increases in respiration are due to increases in mitochondrial content or intrinsic changes in mitochondrial function. Interestingly, the fundamental observation of increased respiratory flux rates in tumors compared to benign tissues was conserved independent of the normalization method, suggesting that HER2-driven mammary tumors are characterized by high respiratory flux rates independent of total cellular protein or reduced mitochondrial protein content. While greater intrinsic OXPHOS was observed with all substrates, lipid-supported respiration was stimulated to a greater extent in the tumors, resulting in comparable carbohydrate (pyruvate, malate, ADP) and lipid (palmitoyl-carnitine, malate, ADP) supported respiration (*Figure 8A*). This effect occurred despite many of the genes and proteins involved in lipid metabolism at the plasma membrane and in the cytosol (e.g. *Acsl1*, *Adipoq*, *Dgat1/2*, *Lipe*, *Fabp4*, *Cd36*, *Pparg*) and mitochondria (e.g. *Etfa*, *Etfb*, *Etfdh*, *Hadh*, *Hadhb*, *Cpt1b*, *Cpt2*) being downregulated in tumors compared to benign mammary tissue (*Figure 8B–E*). The intrinsic activation of respiration was not a ubiquitous finding for all processes linked to OXPHOS, as reactive oxygen species (*Figure 8—figure supplement 1A*) emission rates were higher in tumors compared to benign tissue when normalized to wet weight (*Figure 8—figure supplement 1B*), lower when normalized to total protein (*Figure 8—figure supplement 1C*), and not different when normalized to mitochondrial content (*Figure 8—figure supplement 1D*).

## High respiratory flux rates cannot be explained by OXPHOS uncoupling

Typically, respiratory capacity correlates with mitochondrial content in fully differentiated tissues (*Boykov et al., 2023*). However, given the apparent discrepancy between mitochondrial/OXPHOS protein content and respiratory flux rates in HER2-driven tumors, we investigated alternative explanations for this observation. One possible explanation for the observed enhanced respiratory rates could be due to the uncoupling of OXPHOS from ATP synthesis, especially when ROS emission is not proportional to respiratory flux. To address this possibility, we utilized oligomycin (OMY) to inhibit the $F_0$ subunit of ATP synthase (Complex V) and examine its inhibitory potency between groups (*Figure 8F*). While OMY inhibited respiration in benign mammary tissue by ~55%, respiration in tumors was inhibited by ~80% (*Figure 8G*). Subsequent additions of antimycin A (AMA) to inhibit Complex III and rotenone (ROT) to inhibit Complex I fully inhibited respiration in both tissues (*Figure 8G*). Since respiration was inhibited to a greater extent in the tumors in response to OMY, this protocol strongly suggests that HER2-driven tumor mitochondria are highly coupled.

## Tumor mitochondria display altered morphology

When assessing the subcellular ultrastructure of HER2-driven mammary tumors using transmission electron microscopy (TEM), we observed smaller, more punctate mitochondria in tumors compared to benign tissue (*Figure 9A*). To determine whether ultrastructural observations aligned with the expression of genes and proteins involved in mitochondrial turnover, we filtered our transcriptomic and proteomic dataset using MitoCarta, specifically investigating mitochondrial proteins involved in mitochondrial dynamics and mitophagy. This approach revealed the upregulation of several genes and proteins involved in mitochondrial fission and generally a downregulation of genes and proteins involved in mitochondrial fusion (*Figure 9B–H*). Together, these data show morphological differences in tumor versus benign tissue mitochondria, which could potentially influence OXPHOS efficiency.

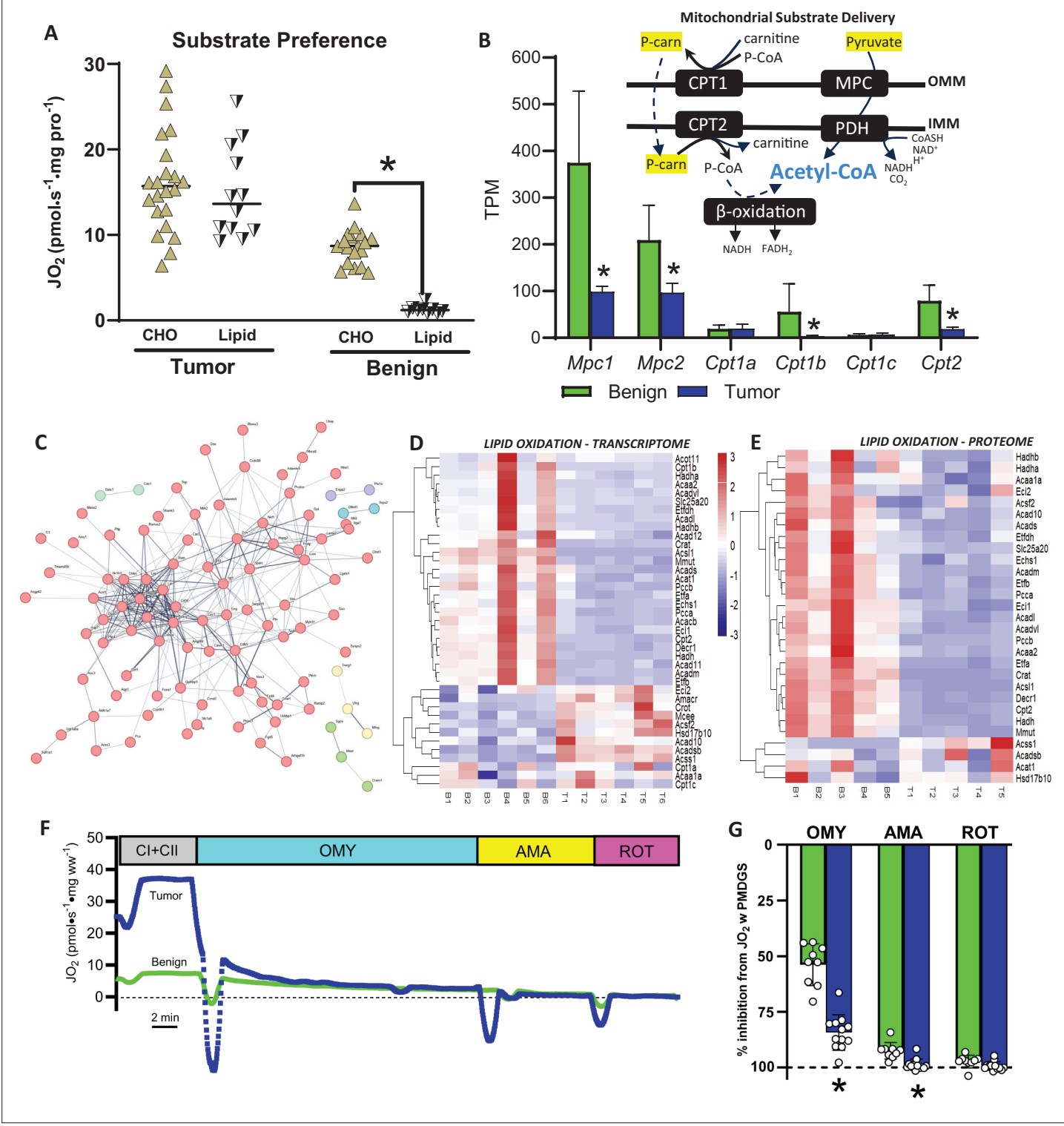

**Figure 8.** Lipid and carbohydrate linked respiration and lipid metabolism transcriptome and proteome. (**A**) Rates of carbohydrate (pyruvate, malate, ADP) and lipid (palmitoyl-carnitine, malate, ADP) supported respiration in tumors and benign mammary tissue. (**B**) Gene expression (transcripts per million) of mitochondrial pyruvate carrier and carnitine palmitoyltransferase isoforms mediating mitochondrial pyruvate and long chain fatty acid transport, respectively. (**C**) Network interaction showing downregulated (unsupervised analysis) gene clustering in the HER2-driven tumor transcriptome. (**D**) Heatmap of differentially expressed genes involved in lipid oxidation. (**E**) Heatmap of differentially expressed proteins involved in lipid metabolism. (**F**) Representative trace of protocol used to assess mitochondrial coupling. Following maximal rates of Complex I- and II-supported respiration, 1 µM oligomycin (OMY) was added to inhibit Complex V (ATP synthase), followed by 2.5 µM antimycin A (AMA) and 1 µM rotenone (ROT) to inhibit complexes

*Figure 8 continued on next page*

*Figure 8 continued*

III and I, respectively. (**G**) Group data for experiment depicted in (**F**) depicted as relative inhibition from maximal Complex I- and II-supported respiration (PMDGS) in each group. In panels **A, B, F, and G** data are presented as means ± SD and statistical analyses were conducted using an unpaired, two-tailed Student's *t*-test (*p < 0.05).

The online version of this article includes the following figure supplement(s) for figure 8:

**Figure supplement 1.** Reactive oxygen species emission in benign mammary tissue and HER2-driven mammary tumors.

## HER2 inhibition suppresses tumor respiration

Although the exact mechanism(s) driving OXPHOS activation despite reductions in mitochondrial content in our model remain unclear, we next asked whether HER2 activity directly regulates mitochondrial bioenergetics. To address this, we used NF639 cells – an epithelial line derived from mammary tumors from MMTV-c-neu transgenic mice, which, like our MMTV/neu$^{ndl}$-YD5 model, display upregulated murine HER2/neu signaling (*Amundadottir and Leder, 1998*; *Muller et al., 1988*; *Min et al., 2007*; *Romieu-Mourez et al., 2002*) serving as a highly relevant cell-line analog to our intact tumor measurements. NF639 cells were treated with lapatinib, a dual HER2/EGFR tyrosine kinase inhibitor (*Figure 10A*), a well-described and validated inhibitor of HER2-mediated signaling (*Serra et al., 2011*; *Xiang et al., 2019*; *Leung et al., 2015*). Cells were exposed to 250 nM lapatinib for 24 hr, a dose and time regimen that was validated and did not notably compromise viability (*Figure 10B*). Using the same CI + CII linked respirometry protocol applied to the murine tumors (*Figure 7B*), lapatinib significantly reduced leak respiration, CI-linked respiration, and CI + CII-supported respiration relative to DMSO-treated cells (*Figure 10C*). Importantly, lapatinib did not induce a cytochrome c (CC) response (% change in respiration with CC, CON = −0.83%, LAP = −2.9%) and did not alter respiratory inhibition with OMY, AMA, or ROT (*Figure 10C*), indicating that mitochondrial integrity and coupling were maintained. Collectively, these findings demonstrate that upregulated HER2-signaling directly stimulates mitochondrial bioenergetics.

## Discussion

The aim of the current study was to investigate the metabolic characteristics of a mouse model of HER2-driven mammary cancer compared to benign mammary tissue. Specifically, we focused on the bioenergetic adaptations that have occurred with HER2-driven tumorigenesis both with respect to mitochondrial function (respiration and ROS) and mitochondrial content (total mitochondrial proteome and OXPHOS complexes). This investigation revealed several findings: (1) HER2-driven mammary tumors exhibit high respiratory rates compared to benign mammary tissue, (2) tumors are capable of oxidizing lipids and pyruvate to a similar extent despite the downregulation of several genes involved in lipid catabolism, (3) high tumor respiration rates cannot be explained by higher mitochondrial content, and (4) HER2 signaling regulates the upregulation of tumor respiration. These data highlight that HER2-driven mammary tumors exhibit high OXPHOS rates despite lower mitochondrial content, challenging the typical bioenergetic relationship in fully differentiated tissues.

### Relationship between mitochondrial content and function in breast cancer

Studies simultaneously assessing mitochondrial content and mitochondrial respiration, and the contribution of these factors to breast cancer malignant transformation are scarce. While reductions in ETC subunit content/activity have been described in mitochondria isolated from human mammary carcinoma cultures compared to mammary epithelial cells (*Putignani et al., 2008*), it has been assumed that a reduction in ETC subunit content would manifest as a reduction in respiratory capacity. However, this is at odds with several reports that classify breast cancer cells as OXPHOS reliant due to increases in mitochondrial respiration correlating with advanced disease (*Rohlenova et al., 2017*; *Hu et al., 2020*; *Pacheco-Velázquez et al., 2018*). Indeed, across many cancers, it is generally assumed that mitochondrial content and respiratory function increase or decrease concomitantly; however, recent work has challenged this dogma, demonstrating that high rates of respiration normalized to tissue weight can be negated when corrected for mitochondrial mass (*Nelson et al., 2021*; *McLaughlin et al., 2020*), revealing intrinsic alterations in cancer cell bioenergetics. In the current study, utilizing this

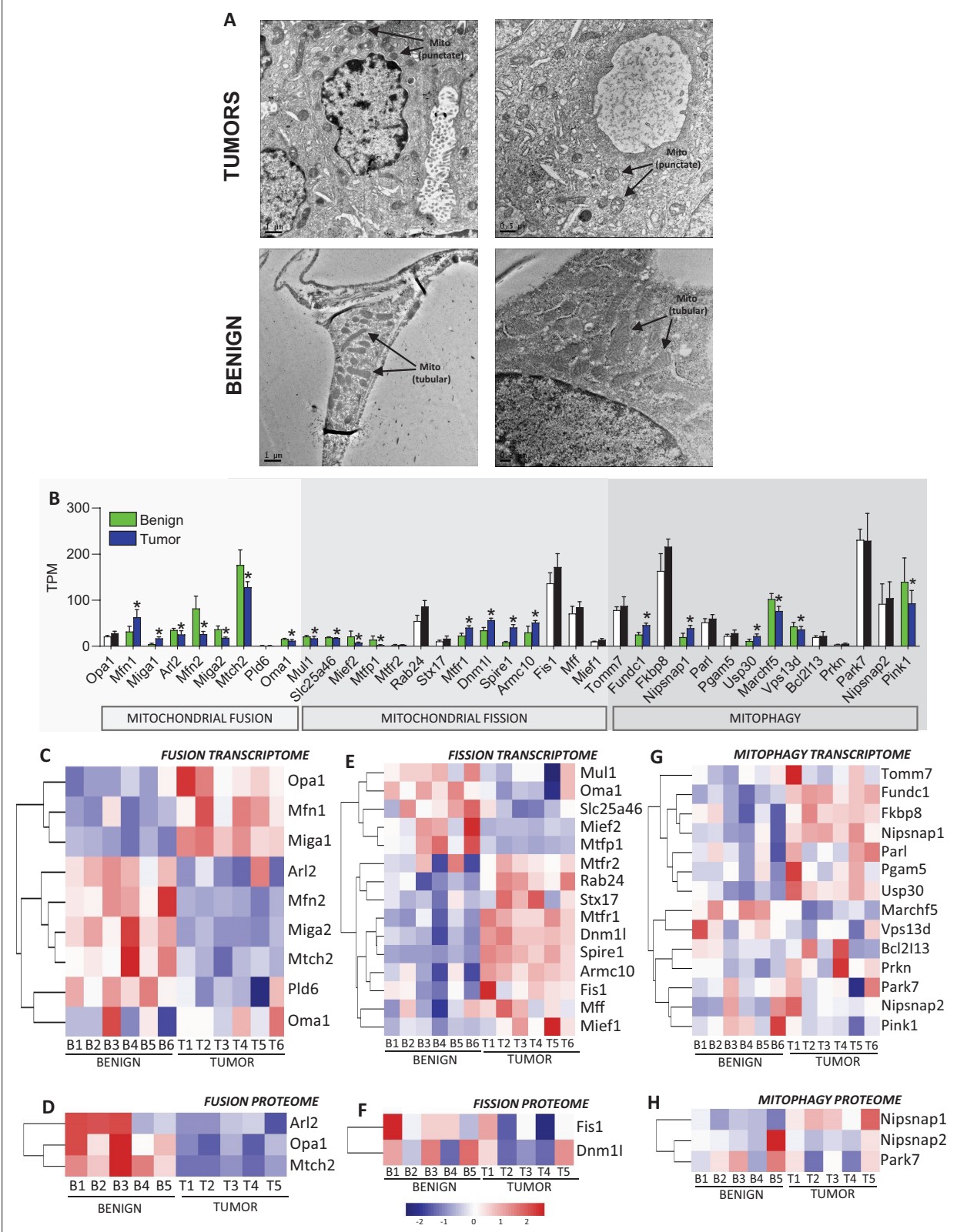

**Figure 9.** Mitochondrial morphology and assessment of mitochondrial fusion, mitochondrial fission, and mitophagy transcriptomes and proteomes in benign mammary tissue and HER2-driven tumors. (**A**) Transmission electron microscopy (TEM) images of tumor and benign samples showing punctate (tumor) versus elongated (benign) mitochondria. (**B**) Gene expression (detected by transcriptomics; transcripts per million) of fusion, fission, and mitophagy related genes separated per process filtered by MitoCarta, colored bars and * represent genes with an adjusted p-value of <0.1. Black and

*Figure 9 continued on next page*

*Figure 9 continued*

white bars represent genes with no significant differences between groups. (**C**) Heatmap of fusion transcriptome. (**D**) Heatmap of fusion proteome. (**E**) Heatmap of fission transcriptome. (**F**) Heatmap of fission proteome. (**G**) Heatmap of mitophagy transcriptome. (**H**) Heatmap of mitophagy proteome. TEM images are from independent tumor and benign tissue samples. Heatmaps depict all detected proteins (*n* = 5 biological replicates/group) and transcripts (*n* = 6 biological replicates/group).

same approach to control for differences in mitochondrial content, we found that respiratory capacity supported by pyruvate and lipids in HER2-driven mammary tumors was significantly higher (~2- to 4-fold) compared to benign mammary tissue despite lower mitochondrial enrichment (total mitochondrial proteome) and lower abundance of several respiratory complex subunits, suggesting mitochondrial content-independent increases in mitochondrial respiratory capacity are acquired during HER2-driven tumorigenesis. This is a key finding independent of the specific mechanisms underlying the enhanced bioenergetics.

While the exact mechanisms underlying the activation of OXPHOS in the current work remain unclear, our findings allow us to speculate on several possibilities that could enhance respiration in a mitochondrial content-independent manner. It is unlikely these factors are mutually exclusive and include increased coupling of electron transfer to ATP synthesis (coupling efficiency), changes in the delivery of substrates across the mitochondrial membranes, intrinsic changes in the activity of ETC complexes and/or ATP synthase, alterations in mitochondrial morphology/dynamics, and signaling events linked to HER2 activation.

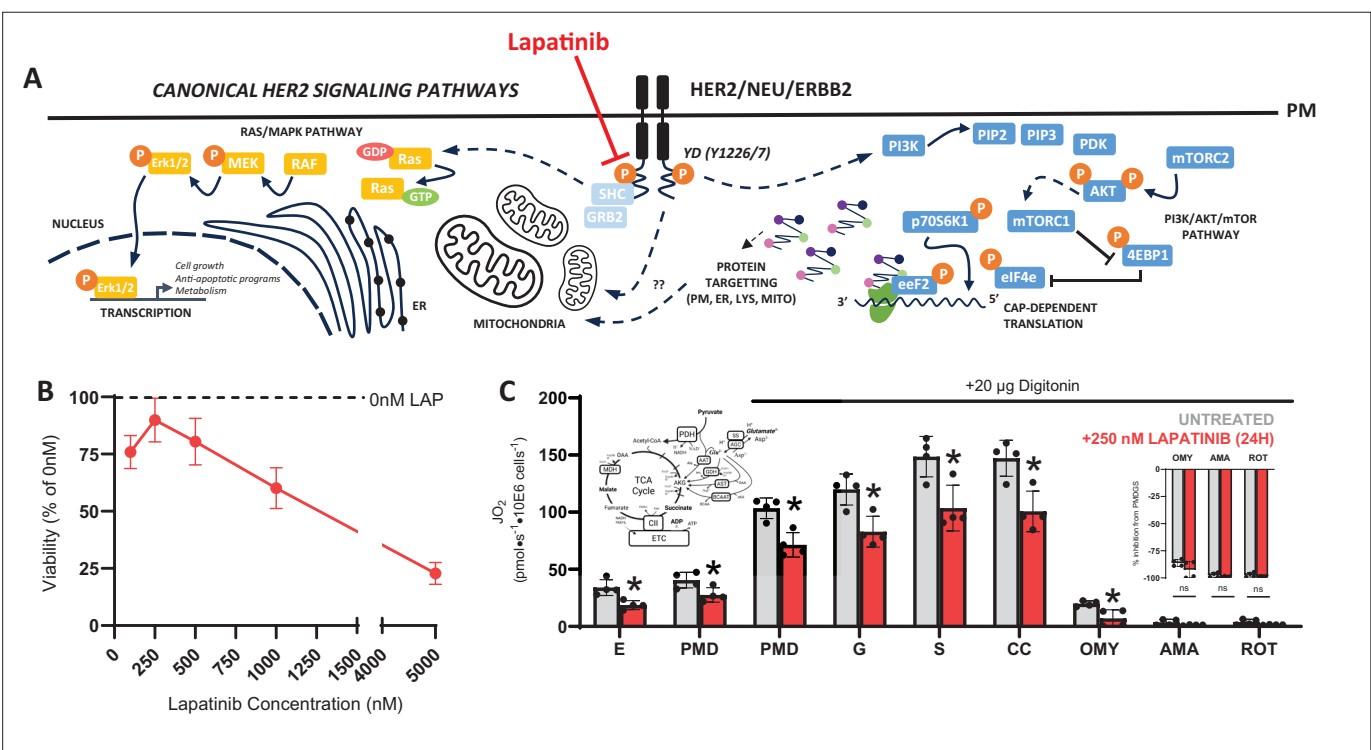

**Figure 10.** Mitochondrial respiration is suppressed by tyrosine kinase inhibitor lapatinib. (**A**) Schematic showing mechanism of lapatinib mediated inhibition of HER2 tyrosine kinase activity. (**B**) NF639 cell viability assessed using a crystal violet assay expressed as a percentage of untreated viability (0 nM lapatinib). (**C**) Complex I- and II-supported respiration in the presence of 20 µg digitonin in NF639 cells untreated (gray) or treated with 250 nM lapatinib for 24 hr. Left inset: Schematic of substrates supporting Complex I–II respiration, schematic of lapatinib or DMSO treatment. Right inset: Percent inhibition of maximal CI + II supported respiration (PMDGSC) with oligomycin (OMY), antimycin A (AMA), and rotenone (ROT). Following maximal rates of Complex I- and II-supported respiration, 1 µM OMY was added to inhibit Complex V (ATP synthase), followed by 2.5 µM AMA and 1 µM ROT to inhibit Complexes III and I, respectively. Each datapoint represents a technical replicate (duplicate), and data are presented as means ± SD. Statistical analyses were conducted using an unpaired, two-tailed Student's *t*-test (*p < 0.05). *Abbrev: E = Endogenous, PMD = pyruvate (5 mM), malate (2 mM), D = ADP (5 mM), G = glutamate (10 mM), S = succinate (10 mM), CC = cytochrome c (10 µM).*

# Possible mitochondrial content-independent mechanisms supporting enhanced respiration rates in HER2-driven mammary cancer

## Coupling efficiency

Several studies have suggested that a key function of mitochondrial respiration in cancer cells is the regeneration of NAD$^+$, FAD, and ubiquinone to provide the necessary metabolites for biomass production, rather than ATP synthesis (*Sullivan et al., 2015*; *Martínez-Reyes et al., 2020*; *Luengo et al., 2021*). From a bioenergetic perspective, this could be achieved by inducing uncoupled respiration, which could manifest in an apparent increase in intrinsic mitochondrial respiration, given that measures of respiratory flux quantify oxygen consumption rather than ATP synthesis. However, in the present study, the Complex V inhibitor OMY inhibited respiration to a greater extent in HER2-driven tumors, suggesting that uncoupling cannot explain the enhanced respiration rates observed, and if anything, tumors are more coupled than benign mammary tissue. This interpretation aligns with work in MCF7 cells where mitochondrial ATP represents the majority of cellular ATP production (*Giddings et al., 2021*), suggesting the ATP synthetic function of OXPHOS in breast cancer is likely intact. Indeed, enhanced mitochondrial ATP provision and OXPHOS-dependent biomass expansion are not mutually exclusive processes, and increased coupled respiration would satisfy both the bioenergetic and biosynthetic requirements of rapid cell proliferation during tumorigenesis.

## Substrate delivery and preference

Recent work has highlighted the dependency of HER2+ tumor cells on both lipid (long chain fatty acids) and carbohydrate-derived (pyruvate) substrate for optimal proliferative capacity (*Nandi et al., 2024*). In addition to pyruvate, HER2-driven tumors may be particularly poised to utilize lipids, given the proximity of cancer cells to adipocytes in the mammary tumor microenvironment (*Wang et al., 2017*), which has spurred interest in inhibiting lipid use/fat oxidation as an adjunct therapy in HER2-driven cancer (*Nandi et al., 2024*). Indeed, the use of endogenous substrate may be preferred as solid tumors expand and factors such as increased interstitial fluid pressure and extracellular matrix remodeling impact tumor vessel quality (*Matuszewska et al., 2021*), possibly limiting exogenous substrate delivery. However, at the transcript and protein level, enzymes involved in mitochondrial lipid uptake (CPT1, CPT2) and pyruvate uptake (MPC) were largely downregulated compared to benign tissue, which would limit rather than enhance mitochondrial substrate delivery. Despite this lower abundance of enzymes supporting mitochondrial substrate delivery, mitochondrial respiration supported by pyruvate and lipids was significantly higher in tumors. Considering lipid and carbohydrate oxidation converge within the mitochondrial matrix, both providing acetyl-CoA to the TCA cycle, it is possible that enhanced OXPHOS efficiency lies at a distal site, implicating intrinsic OXPHOS adaptations as a possible explanation for enhanced respiration in HER2-driven tumors.

## Intrinsic OXPHOS adaptations

Several cancers are characterized by the altered expression of individual ETC subunits and/or ATP synthase, which can have functional repercussions on respiratory function (*Smith et al., 2020*; *Nelson et al., 2021*). In the current study, despite a consistent downregulation of subunits comprising Complexes I–IV in HER2-driven tumors, a subset of Complex V genes were not downregulated (specifically two subunits of the c-ring of F$_0$ domain (*Atp5g1*, *Atp5g2*), two subunits of the F6 domain stalk connecting the F$_0$ and F$_1$ domains (*Atp5j1*, *Atp5j2*), and regulatory proteins ATPase inhibitory factor 1 (*Atpif1*) and distal membrane assembly component 2 (*Dmac2l*)). This shift essentially increases the ratio of ATP synthase/ETC and raises the possibility that a maintenance of these subunits may enhance OXPHOS rates in the face of downregulated CI–IV subunit content since maximal coupled respiration is usually limited by proton flux through ATP synthase. Interestingly, this observation was further supported by our assessment of scRNA-seq data in human breast tumors, where CV expression was generally uniformly and highly expressed across major cell types, further supporting the notion that the maintenance of CV may be important for tumorigenesis. Indeed, despite mtDNA lesions in proton pumping complexes (CI, CIII, and CIV) being common in many cancers, presumably reducing the content/function of these complexes, ATP synthase mutations are less common, suggesting positive selection may exist to maintain coupled respiration and ATP production (*Smith et al., 2020*; *Wu et al.,*

*2018*; *Schöpf et al., 2020*). Thus, while speculative, it is possible that higher respiration rates in HER2-driven tumors may partially reflect maintained ATP synthase content/function.

While bioenergetic efficiency is typically related to intrinsic changes in OXPHOS stoichiometry, mitochondrial morphology can indirectly affect efficiency. We observed smaller, punctate mitochondria in HER2-driven mammary tumors when visualized by TEM, generally aligning with reductions in mitochondrial fusion and increases in mitochondrial fission-related targets. While the direct relationship between mitochondrial morphology and bioenergetics remains contentious, evidence suggests smaller mitochondria (due to enhanced fission or reduced fusion) may be associated with enhanced bioenergetics. Smaller mitochondria tend to have a higher surface-area-to-volume ratio, which has been correlated with upregulated OXPHOS in the liver (*Zhou et al., 2022*), and low rates of respiration are associated with the inhibition of mitochondrial fission (*Lhuissier et al., 2024*). Furthermore, rare genetic mitochondrial diseases are often characterized by hyper-fused mitochondria, which are associated with OXPHOS limitations (*Zou et al., 2021*). Indeed, morphological differences in breast cancer mitochondria are well described (*Baek et al., 2023*; *Ren et al., 2021*; *Zhao et al., 2013*; *Han et al., 2015*), and targeting mitochondrial dynamics for therapy is an active area of study (*Zamberlan et al., 2022*; *Yin et al., 2016*; *Tang et al., 2018*); however, there is a clear need to ascertain the relationship between mitochondrial dynamics and bioenergetics in tumorigenesis.

### Intrinsic activation of OXPHOS mediated by HER2

No matter the mechanisms underlying our observed bioenergetic activation in HER2-driven tumors, our experiments in NF639 cells demonstrate that HER2 activation appears to regulate respiratory function. Although findings in the literature regarding how HER2 activation or suppression influences mitochondrial content and bioenergetics remain mixed, several lines of evidence support the idea that HER2 is important for tumorigenesis and respiratory competence (*Rohlenova et al., 2017*; *Novotna et al., 2024*; *Ding et al., 2012*; *Kurmi et al., 2018*). Importantly, these studies collectively reinforce that changes in mitochondrial content alone cannot fully account for alterations in respiratory function, suggesting that HER2 may regulate intrinsic mitochondrial activity. Independent groups have shown that HER2 may localize to the inner mitochondrial membrane and directly interact with respiratory complexes (CI, CIV, CV), influencing their activity (*Rohlenova et al., 2017*; *Novotna et al., 2024*; *Ding et al., 2012*). Alternatively, indirect activation of bioenergetics via HER2 signaling is another possibility, which is supported by work showing mitochondrial ADP availability may be enhanced via mitochondrial creatine kinase (MtCK) phosphorylation by HER2 downstream effectors (*Kurmi et al., 2018*). Both mechanisms would alter bioenergetics without necessarily altering mitochondrial content, which aligns with our observation that content markers do not scale with respiratory capacity. Nonetheless, further research is required to more clearly delineate the relationship between HER2 signaling, OXPHOS activation, and possible mitochondrial content-independent mechanisms conferring respiratory competence.

### Conclusion

We provide evidence that HER2-driven mammary tumors display enhanced respiratory capacity despite lower mitochondrial content. This increased respiratory capacity could not be explained by changes in OXPHOS coupling; however, differences in mitochondrial morphology and the maintained expression of several subunits of ATP synthase suggest HER2-driven mammary tumors may undergo intrinsic adaptations in mitochondrial function during tumorigenesis. No matter the mechanism driving OXPHOS activation in our model, upregulated HER2 signaling is essential for the observed increases in mitochondrial respiratory capacity.

## Materials and methods
### Animals and breeding

All rodent experiments were conducted in accordance with institutional guidelines approved by the Animal Care Committee at the University of Guelph (AUP#5071). MMTV/*neu*$^{ndl}$-YD5 mice on an FVB background were obtained from Dr. William Muller (McGill University) and Dr. David Ma (University of Guelph). Female MMTV/*neu*$^{ndl}$-YD5 mice were obtained by breeding male heterozygous MMTV/*neu*$^{ndl}$-YD5 mice with female FVB mice to yield WT or heterozygous transgenics. Mice were group housed

in ventilated cages at room temperature (22°C) and kept on a 12:12 light–dark cycle with ad libitum access to a chow diet and water. Starting at 10 weeks, mice were monitored for tumor development on a biweekly basis for ~10 additional weeks. Mice with tumors exceeding 17 mm in length/width or a volume over 5000 mm$^3$ were euthanized before the 20-week time point.

### Tumor and benign mammary tissue collection

Mice were anesthetized using sodium pentobarbital (60 mg/kg body weight) and checked for depth of anesthesia by leg retraction after tail pinch. Once under the surgical plane, a ventral incision was made up the midline and the skin separated bilaterally from the underlying fascia to allow for the excision of the intact mammary tumors. Depending on the number of tumors present (typically 1–5), tumors were allocated for mitochondrial function (placed fresh in BIOPS preservation buffer), histology (TEM: fixed in 2.5% glutaraldehyde/1% paraformaldehyde, IF: fixed in 10% formalin), or snap frozen in liquid nitrogen for further analysis (western blotting, transcriptomics, quantitative label-free proteomics) immediately after whole tumor wet weight was recorded. Paired mammary tissue with no visible tumors, furthest from the site of tumor excision (i.e. if tumors localized in cervical region benign tissue was harvested from inguinal region) were used as an internal comparison, denoted as benign mammary tissue. While mitochondrial respiration experiments were performed in paired tissue (benign and tumor from same mouse), transcriptomic, proteomic, and histological analyses were performed on tumors and benign samples from different mice due to tissue limitations.

### RNA sequencing and transcriptomic analyses

Frozen tissues were homogenized and lysed in TRIzol reagent. After centrifuging, the supernatant (aqueous phase) was applied to the RNeasy kit (QIAGEN, 74106) for subsequent total RNA extraction and purification in accordance with the manufacturer's protocol. RNA samples were sent to the McMaster Genomics Facility where an RNA quality check, poly A enrichment, library prep, and quality check were performed as previously described (*Wang et al., 2023*). These samples were then run in an Illumina Nextseq P2, 2 × 50 bp sequencing run. FastQC and MultiQC were used for quality control of raw data. Salmon's transcript-level quantification (*Patro et al., 2017*) and DESeq2 (*Love et al., 2014*) was used to detect DEGs with the threshold of adjusted p-value <0.1. PCA analysis was performed by using variance stabilizing transformation (VST) data through DESeq2 using the pcaExplorer package (*Marini and Binder, 2019*) in R (version 4.2.1). Functional enrichment analysis was performed by GO enrichment analysis (*Mi et al., 2019*) and KEGG mapping using the GOstats, KEGG.db, and Bioconductor packages. Heatmaps and volcano plots were generated with ggplot2 package using all detected genes or genes sorted by an adjusted p-value <0.1 by false discovery rate method. Detailed descriptions for each analysis are included in the figure descriptions.

### Western blotting

Tumor and benign mammary tissues were homogenized in lysis buffer, diluted to equal protein concentration (1 µg/µl), separated by SDS–PAGE, transferred onto PVDF membranes, and detected using enhanced chemiluminescence (ChemiGenius2 Bioimaging System, SynGene, Cambridge, UK) as previously described (*Paglialunga et al., 2015*; *Miotto and Holloway, 2019*). Commercially available primary antibodies were used to detect total and phosphorylated targets (p38 MAPK (CS9212S, 1:1000), ERK1/2 (CS9102S, 1:1000), Akt (CS4691S, 1:1000), mTOR (CS2972S, 1:1000), p70s6k1 (CS9202S, 1:1000), eEf2 (CS2332S, 1:1000), p-p38 MAPK (CS9211S, 1:1000), p-ERK1/2 (CS9101S, 1:1000), p-AKT (CS9271S, 1:1000), p-mTOR (CS2971S, 1:1000), p-p70s6k1 (CS9234S, 1:1000), p-eEF2 (CS2331S, 1:1000), OXPHOS subunits SDHB (of CII) and MTCO1 (of CIV) (Ab110413, 1:1000)). Due to rapid tissue/protein turnover in tumor tissues, Ponceau staining was an inappropriate loading control; however, trends in total protein targets between groups aligned with quantitative label-free proteomic analyses as a secondary method measuring protein content, validating targets detected by western blotting.

### Protein extraction and purification for LC–MS

A Bradford assay was used to determine the protein concentration in the samples. Twenty-five µg of protein per lysate was resolubilized with denaturation buffer (6 M urea/2 M thiourea) and concurrently reduced with 10 mM DTT and alkylated with 20 mM iodoacetamide at RT for 60 min. Samples were

precipitated by adding 6:1 vol/vol cold acetone and kept at –80°C for 60 min and the pellet was collected after centrifugation at 10,000 × $g$ for 10 min at 4°C. Samples were resuspended with 50 mM ammonium bicarbonate, and MD-grade trypsin (Thermo Fisher Scientific, Cat#90057) was added at a ratio of 1:50 protease to protein. Digestion occurred overnight at 37°C. After digestion, samples were dried by vacuum centrifugation and purified using Pierce C18 Spin columns (Thermo Fisher Scientific, cat#89873).

## LC–MS

The Vanquish Neo UHPLC system was coupled with Orbitrap Exploris 240 mass-spectrometer using the Easy-Spray source for nanoLC-MS protein identification. The Vanquish Neo UHPLC system was configured for trap and elute analysis. Peptides were first trapped and washed on a Pepmap Neo C18 trap column (5 μm, 300 μm × 5 mm) then separated on EASY-Spray columns 75 μm I.D. × 50 cm with the maximum pressure of 1200 bar. The nanoLC-MS system was controlled with Standard Instrument Integration (SII) for Xcalibur software. All hardware and data acquisition software were from Thermo Fisher Scientific. The mobile phase A and weak wash liquid was water with 0.1% FA, and the mobile phase B and strong wash liquid was 80% acetonitrile with 0.1% FA. The gradient was as follows: 4–19% B over 72 min, 19–29% B over 28 min, 29–45% B over 20 min, and a 14.5 min wash at 100% B with a flow rate of 300 nl/min. The autosampler temperature was 7°C, and the column temperature was 45°C. The sample was injected with Fast Loading set to 'Enabled' with Pressure Control at 500 bar. The column Fast Equilibration function was set to 'Enabled' with Pressure control at 800 bar, and the equilibration factor was set to 3. Vial bottom detection was set to 'Enabled'.

## Data-dependent acquisition MS method

The Orbitrap Exploris 240 MS was operated in data-dependent acquisition (DDA) mode using a full scan with $m/z$ range 375–1500, Orbitrap resolution of 60 000, normalized AGC target value 300%, and maximum injection time set to Auto. The intensity threshold for precursor was set to $1 \times 10^4$. MS/MS spectra starting from 120 $m/z$ were acquired in DDA mode with a cycle time of 2 s, where the precursors were isolated in a window of 1.6 Da and subsequently fragmented with HCD using a normalized collision energy of 30%. Orbitrap resolution was set to 15,000 for MS2. The normalized AGC target was standard, and the maximum injection time was set to Auto.

## Proteomics data analysis

Analysis of the tissue proteome was performed in R (version 4.2.1) and R Studio (version 2024.04.2+764). PCA was performed using the Vegan package. Volcano plots were created using log2 transformed fold change comparing cancer/benign tissue and -log10 adjusted p-value. p-values of differentially regulated proteins between each group were corrected with p.adjust (method = 'fdr') within each comparison. Depending on the analysis, heatmaps and volcano plots were generated using all detected proteins or proteins sorted by p.adjust <0.1; detailed descriptions for each analysis are included in the figure descriptions. Gene set enrichment was performed with the KEGG (*Kanehisa et al., 2016*) and EnrichR (*Xie et al., 2021*).

## Cell culture

NF639 murine mammary epithelial tumor cells were obtained directly from ATCC (CRL-3090, Lot #70017140) and used at p14 across all experiments. ATCC performs comprehensive authentication and quality-control testing on all distribution lots, and all cells tested negative for mycoplasma contamination (ATCC certified). Cell morphology was additionally monitored regularly by light microscopy to verify identity, and passage number was tracked consistently across experiments to ensure reproducibility. NF639 does not appear on the ICLAC Register of Misidentified Cell Lines. NF639 cells were cultured in DMEM supplemented with 10% fetal bovine serum (FBS; Gibco), 4 mM L-glutamine, 1 mM sodium pyruvate, 50 U/ml penicillin, and 50 μg/ml streptomycin (Wisent), maintained by passaging in 100 mm plates, and incubated at 37°C and 5% $CO_2$. Cells were seeded in 96-well plates ($10^4$ cells/well; $n$ = 3 technical replicates) for microplate viability assays or in 100 mm plates for mitochondrial respiration experiments. At 60% confluency, cells were serum-starved for 6 hr, followed by lapatinib treatment (Cayman, 11493) in complete culture medium containing 2% FBS for 24 hr. Cell viability in the presence of lapatinib was tested using a crystal violet assay. Cells were treated with lapatinib at

0, 100, 250, 500, 1000, or 5000 nM for 24 hr. Following treatment incubation, cells were washed and stained with crystal violet dye (Abcam), and absorbance was measured at 570 nm as per the manufacturer's instructions. Assays were repeated in triplicate. For mitochondrial respiration experiments, cells were treated with 250 nM lapatinib or DMSO (control) in complete culture medium containing 2% FBS for 24 hr. Cells were washed, trypsinized (1x trypsin-EDTA, Wisent), and centrifuged at 300 × $g$ for 5 min before counting. Cells were washed and resuspended in 1x PBS and centrifuged at 300 × $g$ for 5 min. Cells were resuspended in Mir05 respiration buffer (0.5 mM EGTA, 3 mM MgCl$_2$·H$_2$O, 60 mM potassium lactobionate, 10 mM KH$_2$PO$_4$, 20 mM HEPES, 110 mM sucrose, 20 mM taurine, and 1 g/l fatty acid free BSA; pH 7.1) for mitochondrial respiration.

## Mitochondrial respiration

Mitochondrial respiration experiments were performed using the Oroboros Oxygraph-2k systems at 37°C in MiR05 respiration with constant stirring at room air saturation (starting at 180 µM O$_2$). For respiration experiments in intact tumors and benign tissue, 2–5 mg of permeabilized tumor tissue or 15–20 mg of permeabilized mammary tissue (both permeabilized with 20 µg/µl saponin) were used per 2 ml chamber, and each protocol was conducted as at least two technical replicates (duplicate) per tumor or benign sample. Prior to the addition of substrates for any protocol, endogenous (basal) respiration was assessed with no substrates added. Complex I- and II-supported respiration was assessed with the addition of 5 mM pyruvate, 2 mM malate, 5 mM ADP, 10 mM glutamate, and 10 mM succinate. This respiratory capacity measurement represents supraphysiological substrate concentrations delivered in the absence of diffusion limitations. In a subset of these experiments, subsequent additions of OMY (1 µM), AMA (2.5 µM), and ROT (1 µM) were added to assess OXPHOS coupling. In a separate protocol, lipid-supported respiration was assessed with 20 µM palmitoyl-carnitine in the presence of 2 mM malate and 5 mM ADP. For respiration experiments in NF639 cells, the same respiration buffers and chamber conditions were utilized. One million cells were added per technical replicate, and Complex I- and II-supported respiration was assessed using the same protocol described in intact tumors and benign tissue in the presence of 20 µg digitonin to ensure permeabilization. Respiration was expressed per viable cell.

## Reactive oxygen species emission

Mitochondrial hydrogen peroxide (H$_2$O$_2$) emission was determined fluorometrically (Lumina, Thermo Scientific, Waltham, MA) in permeabilized mammary tissue and mammary tumors as previously described (*Paglialunga et al., 2015*) with minor modifications. In brief, 2–5 mg of minced tissue was loaded into a cuvette containing Amplex Red (Invitrogen, Waltham, MA, USA), horseradish peroxidase (5 U/ml), saponin (20 µg/µl), superoxide dismutase (40 U/ml) in Buffer Z (105 mM K-MES, 30 mM KCl, 1 mM EGTA, 10 mM K$_2$HPO$_4$, 5 mM MgCl$_2$, 5 µM glutamate, 5 µM malate, 0.5% BSA, pH 7.4). Emission rates were assessed at 37°C in the presence of 10 mM succinate and calculated using a standard curve generated with known concentrations of H$_2$O$_2$.

## Data normalization

Mitochondrial respiration rates and H$_2$O$_2$ emission were normalized to individual sample wet weight, group total protein content, and group mitochondrial content (MEF). Total protein content was calculated as a group average ($n$ = 6 biological replicates/group) based on a Bradford assay and back-calculated total protein in a given wet weight of starting tissue. Data was normalized to mitochondrial protein by calculating the MEF for each group ($n$ = 6 biological replicates/group) based on all detected mitochondrial proteins by quantitative label-free proteomics (Mouse MitoCarta 3.0 database) normalized to total protein abundance per sample as previously described by others (*Boykov et al., 2023*; *McLaughlin et al., 2020*).

## Histology

### Transmission electron microscopy

Immediately following tissue dissection, tumor and benign mammary tissues were fixed in 2.5% glutaraldehyde/1% paraformaldehyde overnight at 4°C. The following day, tissues were washed 3x in 100 mM HEPES buffer in preparation for embedding. Tissue was stained with 1% osmium tetroxide for 3 hr, washed 3x in HEPES buffer, and further stained with 1% uranyl acetate for 3 hr before washing

2x in HEPES buffer and once in ddH2O. Stained tissue was dehydrated with a graded ethanol series, followed by infiltration with LR White resin. Blocks were polymerized at 60°C for 18 hr before ultra-microtomy. Sections were post-stained with uranyl acetate and lead citrate. Sections were allowed to fully dry before imaging on the FEI Tecnai G2 F20 transmission electron microscope operated under normal conditions at 120 kV.

## Immunofluorescence

Tumor and benign samples ($n$ = 4 biological replicates/group) were fixed in 10% formalin for 24 hr and stored in 70% ethanol at 4°C. Samples were blocked in paraffin and sectioned, deparaffinized in xylene, and rehydrated through a graded ethanol series. Samples were washed in PBS and incubated for 20 min in 0.02% sodium borohydride in PBS to quench endogenous phosphatase activity. Antigen retrieval was performed in citrate buffer (90°C) for 12 min, followed by a 20 min cooling period. Sections were then blocked in 5% BSA in PBS for 1 hr to prevent nonspecific antibody binding. Primary antibodies (anti-vimentin (1:100, Novus Bio NBP1-31327), anti-cytokeratin (1:100, Novus Bio NBP2-29429), anti-CD4 (1:200, Abcam AB183685), anti-CD138 (1:100, StemCell Technologies 60035), anti-CD31 (1:150, Abcam AB28364), or anti-CD68 (1:100, Novus Bio NB100-683)) were diluted in 1% BSA and applied overnight at 4°C in a humidified chamber. The following day, sections were incubated for 2 hr at room temperature with the appropriate fluorophore-conjugated secondary antibodies (Invitrogen): Alexa Fluor 594 anti-rabbit (1:100 or 1:200), Alexa Fluor 488 anti-mouse (1:100), Alexa Fluor 488 anti-rat (1:200), or Alexa Fluor 488 anti-rabbit (1:100). Slides were washed and mounted with ProLong Gold Antifade Mountant containing DAPI (Thermo Fisher) and protected from light for 24 hr before being transferred to –20°C for storage. Images were acquired at ×40 magnification using an Olympus BX-61 epifluorescence microscope equipped with Metamorph imaging software. Monochromatic images captured at 594 nm excitation were pseudocolored red, while those captured at 488 nm were pseudocolored green. Staining intensity was quantified in ImageJ using manual region-of-interest selection, and immunopositive area was expressed as a percentage of total tissue area (expressed as % immunopositive area).

## OXPHOS index and OXPHOS score calculations

A publicly available single-cell RNA-seq dataset from 26 human breast tumors (*Wu et al., 2021*) was combined with cell-type proportion analyses from our tumor and benign tissues (see Immunofluorescence methods) to estimate cell-type-specific contribution to OXPHOS.

Using human MitoCarta 3.0, we filtered for all detected OXPHOS subunit genes in the scRNA-seq dataset. For each major cell type described in the dataset (B-, T-, plasmablasts, CAFs, PVL, endothelial, myeloid, normal epithelial, and cancer epithelial cells), we extracted the percent of cells expressing each subunit gene. These percent expression values were averaged per complex to generate a mean CI–V expression value for each cell type. To account for differences in cell-type abundance within the scRNA-seq dataset, this percent expression was multiplied by the cell-type proportion reported in the same study. In subsequent calculations, this is reported as *%Expression$_{CT}$*.

To estimate cell-type-specific OXPHOS expression in our benign and tumor samples, we calculated an *OXPHOS mRNA Index* for all cell types present in both datasets (myeloid, T-, plasmablasts, endothelial, mesenchymal, and epithelial cells):

$$OXPHOS\ mRNA\ Index = \left(\%Expression_{CT}\right) * \left(\%Immunopositive_{CT}\right)$$

where %Expression$_{CT}$ is the average gene expression data per respiratory complex in each complementary major cell type (CT), and %Immunopositive$_{CT}$ is cell type proportion measured by IF in our samples. This index represents how much each cell type is expected to contribute to CI–V expression in our tissues, integrating both expression and abundance.

Next, to estimate the contribution of each cell type to the OXPHOS transcriptome in the current study, the OXPHOS mRNA index in each cell type was summed to calculate a Total OXPHOS mRNA Score.

$$Total\ OXPHOS\ mRNA\ Score = \sum_{i=I,II,III,IV,V} OXPHOS\ RNA\ Index\ Ci$$

## Statistical analysis

Statistical analysis and data visualization were performed using Microsoft Excel, GraphPad Prism (version 10.2.3), R (version 4.2.1), and R Studio (version 2024.04.2+764). All data are presented as mean ± SD and data was considered statistically significant if $p < 0.05$. For transcriptomic and proteomic analyses, an adjusted p-value of <0.1 was utilized. As this study was exploratory, formal prospective sample size calculations were not performed; instead, sample sizes were determined based on practical and ethical considerations and are consistent with accepted standards in the field for each experimental platform. Post hoc power estimates based on observed variability and effect sizes are reported here for transparency. For tissue-based mitochondrial respirometry, $n$ = 14–20 biological replicates per group were used, each performed in technical duplicate with reported values representing the mean of technical replicates. Observed biological variability was moderate across substrate states (CV: 23–46%; SD range: 0.29–12.67 across groups and substrates), and the observed differences between benign and tumor tissue were large across all substrate states (Cohen's d range: 2.88–3.52; fold change: 4.9–9.9x), providing >99% power at a two-tailed $\alpha$ = 0.05. For cell culture respirometry, $n$ = 4–5 independent biological replicates per condition were used, each performed in technical duplicate with reported values representing the mean of technical replicates. Observed biological variability was low-to-moderate across primary substrates (CV: 9–20%; SD range: 2.43–17.66), with large effect sizes between treatment conditions (Cohen's $d$ range: 1.97–3.24), providing 80–100% power at a two-tailed $\alpha$ = 0.05. For transcriptomics and quantitative label-free proteomics, $n$ = 5–6 biological replicates per group were used, consistent with accepted standards for discovery-based analyses employing FDR correction across analytes. Details regarding specific statistical analyses are included in the Results section and within each figure legend.

## Acknowledgements

This work is funded by a National Sciences and Engineering Research Council (NSERC) grant by GPH (400362). SMF is funded by an NSERC-CGS doctoral scholarship.

## Additional information

### Funding

| Funder | Grant reference number | Author |
| --- | --- | --- |
| Natural Sciences and Engineering Research Council of Canada | 400362 | Graham P Holloway |

The funders had no role in study design, data collection, and interpretation, or the decision to submit the work for publication.

### Author contributions

Sara M Frangos, Conceptualization, Data curation, Formal analysis, Validation, Investigation, Visualization, Methodology, Writing - original draft, Project administration, Writing – review and editing; Henver S Brunetta, Conceptualization, Data curation, Formal analysis, Investigation, Methodology, Writing – review and editing; Dongdong Wang, Maria Joy Therese Jabile, Grace Mencfeld, Cezar M Khursigara, Investigation, Writing – review and editing; Leslie M Jeffries, Formal analysis, Investigation, Writing – review and editing; David WL Ma, William J Muller, Resources, Writing – review and editing; Kelsey H Fisher-Wellman, Conceptualization, Methodology, Writing – review and editing; Jim Petrik, Conceptualization, Formal analysis, Investigation, Methodology, Writing – review and editing; Gregory R Steinberg, Investigation, Methodology, Writing – review and editing; Graham P Holloway, Conceptualization, Resources, Supervision, Funding acquisition, Methodology, Writing - original draft, Writing – review and editing

### Author ORCIDs

Sara M Frangos ⬚ https://orcid.org/0000-0002-1590-8826

### Ethics

All experiments were performed in accordance with institutional guidelines approved by the Animal Care Committee at the University of Guelph (AUP#: 5071).

Reviewer #1 (Public review): https://doi.org/10.7554/eLife.104079.3.sa1
Reviewer #2 (Public review): https://doi.org/10.7554/eLife.104079.3.sa2
Author response https://doi.org/10.7554/eLife.104079.3.sa3

---

## Additional files

### Supplementary files

Supplementary file 1. List of nine significantly upregulated proteins between tumor and benign samples, sorted in order of fold change from benign (smallest to largest fold change). General pathways or functions are listed per protein based on ID mapping from the Uniprot protein sequence database. Adjusted p-values are listed per target.

MDAR checklist

### Data availability

Raw RNA sequencing and proteomics data were deposited in GEO (accession number: GSE315552) and PRIDE databases (accession number: PXD071897), respectively.

The following datasets were generated:

| Author(s) | Year | Dataset title | Dataset URL | Database and Identifier |
|---|---|---|---|---|
| Frangos S, Brunetta H, Wang D | 2026 | HER2-driven mammary tumorigenesis enhances bioenergetics despite reductions in mitochondrial content | https://www.ncbi.nlm.nih.gov/geo/query/acc.cgi?acc=GSE315552 | NCBI Gene Expression Omnibus, GSE315552 |
| Fragos S, Holloway G | 2026 | MMTV-NEU-NDL-YD5 TUMOR AND BENIGN LCMS | https://www.ebi.ac.uk/pride/archive/projects/PXD071897 | PRIDE, PXD071897 |

The following previously published dataset was used:

| Author(s) | Year | Dataset title | Dataset URL | Database and Identifier |
|---|---|---|---|---|
| Al-Eryani G, Roden D, Junankar S, Harvey K, Andersson A, Thennavan A, Wang C, Torpy J, Bartonicek N, Wang T, Larsson L, Kaczorowski D, Weisenfeld N, Uytingco CR, Chew JG, Bent ZW, Chan C, Gnanasambandapillai V, Dutertre C, Gluch L, Hui MN, Beith J, Parker A, Robbins E, Segara D, Cooper C, Mak C, Chan B, Warrier S, Ginhoux F, Millar E, Powell JE, Williams SR, Liu X, O'Toole S, Lim E, Lundeberg J, Perou CM, Swarbrick A, Wu Sz | 2021 | A single-cell and spatially resolved atlas of human breast cancers | https://www.ncbi.nlm.nih.gov/geo/query/acc.cgi?acc=GSE176078 | NCBI Gene Expression Omnibus, GSE176078 |

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
