## [Editor Report · eLife Assessment]

This **valuable** study aims to determine mechanisms underlying breast cancer initiation and tumour progression. The manuscript includes a **solid** set of transcriptomic and proteomic datasets from tumour samples and examines mitochondrial function within the tumours. While the underlying mechanisms linking expression changes to functional effects remain speculative. This paper provides a resource for researchers working on breast cancer and/or HER2-driven bioenergetics changes.

---

## [Referee Report · Reviewer #1 (Public review)]

Summary:

In this manuscript, Frangos at al. used a transcriptomic and proteomic approach to characterise changes in HER2-driven mammary tumours compared to healthy mammary tissue in mice. They observed that mitochondrial genes, including OXPHOS regulators, were among the most down-regulated genes and proteins in their datasets. Surprisingly, these were associated with higher mitochondrial respiration, in response to a variety of carbon sources. In addition, there seems to be a reduction in mitochondrial fusion and an increase in fission in tumour tissues compared to healthy tissues.

Strengths:

The data are clearly presented and described.

The author reported very similar trends in proteomic and transcriptomic data. Such approaches are essential to have a better understanding of the changes in cancer cell metabolism associated with tumorigenesis.

The authors provided a direct link between HER2 inhibition and OXPHOS, strengthening the mechanistic aspect of the work.

Weaknesses:

The manuscript would have benefited from more ex-vivo approaches to further dissect mechanistic links and resolve the contradiction of elevated respiration with reduced expression of most associated proteins (but these points are clearly articulated in the discussion).

The results presented support the authors' conclusions, and limitations are addressed in the discussion. This work will likely impact the progression of the field, and the provided data will benefit the scientific community.

Comments on revisions:

The authors addressed all my concerns.

---

## [Referee Report · Reviewer #2 (Public review)]

Frangos et al present a set of studies aiming to determine mechanisms underlying initiation and tumour progression. Overall, this work provides some useful datasets, further establishing mitochondrial dysfunction during the cellular transformation process.

A key strength is the coordinated analysis of transcriptomics and proteomics from tumour samples derived from a Neu-dependent mouse model for breast cancer. This analysis provides rigorous datasets that show robust patterns, including down-regulation across many components of mitochondrial OXPHOS that were generally consistent at both the mRNA and protein level. Parallel analysis of corresponding tumour samples thereby clearly shows the opposite trend of increased mitochondrial function, which is unexpected. As such, this work further establishes altered mitochondrial phenotypes in tumour contexts and further illustrates that mitochondrial function is not necessarily always tightly correlated with mitochondrial gene expression patterns.

Several key weaknesses remain. It remains unclear how increased mitochondrial function is being sustained despite wide decreases in mRNA and protein levels of OXPHOS components. In terms of mechanism, the study confirmed that pharmacologic EGFR inhibition decreases OXPHOS in a EGFR-dependent breast cancer line. However, it remains unclear if the cell culture system recapitulates other key observations of the tumour model (namely decreased expression with increased function).

Therefore, the mechanistic basis of increased mitochondrial function in light of decreased mitochondrial content remains speculative, as does the role of these changes for tumour initiation or progression.

Comments on revisions:

We agree with the overall findings of the study and appreciate that the claims in text and title have been appropriately toned down.

As additional suggestions eg for presentation, many of the graphics/labels are still too small to be useful. It would be interesting to see if this cell line is similar to the tumours in terms of all the phenotypes. The lapatinib experiment was good. I wonder how quick this drug affects the mitochondria. Also it would be interesting to see if these cells have higher OXPHOS than other non-transformed breast epithelial cells.

The WB on oxphos components is good with ab110413 but this looks like many subunits are detected so this should be made clear.

---

## [Author Response]

The following is the authors’ response to the current reviews.

**Reviewer #1 (Public review):**
Comments on revisions: The authors addressed all my concerns.

We thank you for the positive review and feedback throughout the review process.

**Reviewer #2 (Public review):**
Comments on revisions: We agree with the overall findings of the study and appreciate that the claims in text and title have been appropriately toned down. As additional suggestions e.g. for presentation, many of the graphics/labels are still too small to be useful. It would be interesting to see if this cell line is similar to the tumours in terms of all the phenotypes. The lapatinib experiment was good. I wonder how quick this drug affects the mitochondria. Also it would be interesting to see if these cells have higher OXPHOS than other non-transformed breast epithelial cells. The WB on oxphos components is good with ab110413 but this looks like many subunits are detected so this should be made clear.

Thank you for these suggestions.

We have clarified in the Methods section (lines 475–476) the specific OXPHOS subunits detected using the Ab110413 antibody cocktail.

With respect to lapatinib, prior work has shown that lapatinib can alter the phosphoproteome within minutes to hours (PMID:22964224). In our experiments, however, NF639 cells were exposed to lapatinib for 24 hours - a timeframe in which transcriptional and translational remodeling are also expected to occur. Therefore, we cannot distinguish whether the observed suppression of OXPHOS reflects acute signaling effects or downstream changes in gene and protein abundance. Importantly, the purpose of this experiment was proof-of-principle: to determine whether HER2 signaling contributes to respiratory competency in a cell line derived from the same transgenic model as the intact tumor slices used in this study. Thus, while defining the precise kinetics of inhibition or comparing to benign/non-transformed cells would be interesting, these were not the primary objectives of the added experiments.

We have increased figure label sizes across all main figures.

The following is the authors’ response to the original reviews.

**Public Reviews:**

**Reviewer #1 (Public review):**
Summary:In this manuscript, Frangos et al. used a transcriptomic and proteomic approach to characterise changes in HER2-driven mammary tumours compared to healthy mammary tissue in mice. They observed that mitochondrial genes, including OXPHOS regulators, were among the most down-regulated genes and proteins in their datasets. Surprisingly, these were associated with higher mitochondrial respiration, in response to a variety of carbon sources. In addition, there seems to be a reduction in mitochondrial fusion and an increase in fission in tumours compared to healthy tissues.Strengths:The data are clearly presented and described.The author reported very similar trends in proteomic and transcriptomic data. Such approaches are essential to have a better understanding of the changes in cancer cell metabolism associated with tumourigenesis.Weaknesses:(1) This study, despite being a useful resource (assuming all the data will be publicly available and not only upon request) is mainly descriptive and correlative and lacks mechanistic links.

We appreciate this point. While the primary goal of our study was to assess mitochondrial adaptations with HER2-driven tumorigenesis, we agree strengthening the mechanistic interpretation would improve the impact of the data. To address this, we have provided experiments demonstrating HER2 inhibition in NF639 cells with lapatinib supresses respiratory capacity, directly supporting the interpretation that HER2 activity regulates respiratory function (Figure 10). We have expanded the discussion appropriately (lines 378-394). Both raw RNA-seq and proteomic data were deposited through GEO and the PRIDE repositories (accession numbers included in Data Availability Statement).

(2) It would be important to determine the cellular composition of the tumour and healthy tissue used. Do the changes described here apply to cancer cells only or do other cell types contribute to this?

We thank the reviewer for this suggestion; we have added experiments that have directly addressed this concern.

Cell type composition analysis by immunofluorescence was added (Figure 6) where we quantified epithelial, mesenchymal, endothelial, immune and stromal populations in our benign mammary tissue and tumor samples. We found no major shift in the dominant cell types that would confound transcriptomic data in whole tissues.

We integrated immunofluorescence data with a publicly available scRNA-seq dataset from human breast tumors which allowed us to estimate cell-type-specific expression of OXPHOS genes in our own samples. Despite the possibility of species differences, this is the only dataset of its kind, and we used this to generate an estimate of cell type weighted OXPHOS mRNA expression (Figure 6). This revealed that epithelial cells are likely the dominant contributors to OXPHOS gene expression for CIIV. All calculations are delineated in the Methods section.

(3) Are the changes in metabolic gene expression a consequence of HER2 signalling activation? Ex-vivo experiments could be performed to perturb this pathway and determine cause-effects.

Thank you for this suggestion – we have included an experiment directly testing this concept. We assessed mitochondrial respiration in NF639 HER2-driven mammary tumor epithelial cells in the presence or absence of the well-described dual tyrosine kinase inhibitor lapatinib. Lapatinib reduced basal, CI-linked and CI+II linked respiration without compromising mitochondrial integrity or coupling, demonstrating that HER2 activation regulates respiration in our model. This data is presented in Figure 10, and a new section has been added to the discussion describing the implications of this finding in the context of the current literature (lines 378-394).

(4) The data of fission/fusion seem quite preliminary and the gene/protein expression changes are not so clear cut to be a convincing explanation that this is the main reason for the increased mitochondria respiration in tumours.

We agree mitochondrial morphology and dynamics alone cannot fully account for the observed respiratory phenotype – this was emphasized in the discussion but has since been further clarified (lines 365-377). We retained the TEM and dynamics gene/protein data because they do support morphological differences consistent with enhanced fission. However, we have revised the tone of our interpretation to more explicitly acknowledge that these findings are correlative, and the updated discussion now emphasizes that the increased respiratory capacity in tumors is likely driven by multiple converging mechanisms.

**Reviewer #2 (Public review):**
Frangos et al present a set of studies aiming to determine mechanisms underlying initiation and tumour progression. Overall, this work provides some useful insights into the involvement of mitochondrial dysfunction during the cellular transformation process. This body of work could be improved in several possible directions to establish more mechanistic connections.(5) The interesting point of the paper: the contrast between suppressed ETC components and activated OXPHOS function is perplexing and should be resolved. It is still unclear if activated mitochondrial function triggers gene down-regulation vs compensatory functional changes (as the title suggests). Have the authors considered reversing the HER2-derived signals e.g. with PI3K-AKT-MTOR or ERK inhibitors to potentially separate the expression vs. functional phenotypes? The root of the OXPHOS component down-regulation should also be traced further, e.g. by probing into levels of core mitochondrial biogenesis factors. Are transcript levels of factors encoded by mtDNA also decreased?

We appreciate this insight and agree that the discordance between mitochondrial content and function is fascinating and have addressed the concerns above in the following manner:

- We have altered the title – we agree we cannot definitively say that the enhanced respiratory capacity observed is compensatory.

- We have added experiments in NF639 cells in the presence of lapatinib, a tyrosine kinase inhibitor to interrogate whether HER2 is necessary for our functional outcome of interest – the enhanced respiratory capacity in the tumors. Lapatinib significantly suppressed respiration (Figure 10) demonstrating HER2 signaling directly regulates mitochondrial respiration.

- We have expanded the discussion to provide further comment on potential explanations for increased respiratory function and low mitochondrial content.

(6) The second interesting aspect of this study is the implication of mitochondrial activation in tumours, despite the downregulation of expression signatures, suggestive of a positive role for mitochondria in this tumour model. To address if this is correlative or causal, have the authors considered testing an OXPHOS inhibitor for suppression of tumorigenesis?

Previous studies have eloquently highlighted that directly or indirectly inhibiting mitochondria can supress growth in HER2-driven breast cancer (PMID:31690671) or alternatively, amplification of mt-HER2 enhances tumorigenesis (PMID: 38291340). In many solid tumors, this is the concept of preclinical and clinical studies using IACS-010759 or similar inhibitors of OXPHOS which do suppress growth but have significant off target effects in healthy tissues (PMID: 36658425, 3580228). We have expanded the discussion to ensure the reader is aware of these previous contributions and highlighted the importance of future work delineating the role of enhanced respiratory function in HER2-driven mammary cancer (lines 378-394).

(7) A number of issues concerning animal/ tumour variability and further pathway dissection could be explored with in vitro approaches. Have the authors considered deriving tumourderived cell cultures, which could enable further confirmations, mechanistic drug studies and additional imaging approaches? Culture systems would allow alternative assessment of mitochondrial function such as Seahorse or flow cytometry (mitochondrial potential and ROS levels).

We thank the reviewer for this suggestion – we have addressed this in part by using the NF639 HER2driven tumor epithelial line which demonstrated that HER2 regulates our observed respiratory response. Unfortunately, the addition of tumor derived cell cultures was not feasible or within the scope of our study. Animal and tumor variability has been clarified in the Methods section (lines 424-429). Mitochondrial respiration experiments were performed in paired tissue (benign and tumor from same mouse). Transcriptomic, proteomic and histological analyses were performed on tumors and benign samples from different mice due to tissue limitations.

(8) The study could be greatly improved with further confirmatory studies, eg immunoblotting for mitochondrial components with parallel blots for phospho-signalling in the same samples. It would be interesting if trends could be maintained in tumour-derived cell cultures. It is notable that OXPHOS protein/transcript changes are more consistent (Figure 5, Supplementary Figure 4) than mitochondrial dynamics /mitophagy factors (Figure 8). Core regulatory factors in these pathways should be confirmed by conventional immunoblotting.

We thank the reviewer for this thoughtful comment. While we agree that additional confirmatory studies can be valuable, due to tissue quantity constraints and the number of assays required for our multi-omics analysis, extensive additional blots were not feasible. However, we had sufficient protein to provide select OXPHOS proteins to verify the proteomic data (now provided in S-Fig.4H). Furthermore, we have plotted the fold change of genes and proteins detected in both datasets and added this to Figure 4 (4A, B), further highlighting the consistency between our transcriptomic and proteomic findings. We believe that the highly consistent and concordant nature of our datasets collectively provides strong support for our central objective - determining whether mitochondrial content and respiratory function correlate in HER2-driven mammary tumors. The reproducibility of OXPHOS-related changes reinforces the robustness of our observations. We also appreciate the reviewer’s insight that OXPHOS alterations appear particularly consistent. In response, we have edited the discussion to further emphasize this point, especially in relation to the distinctive pattern observed for Complex V, which showed greater preservation relative to Complexes I–IV across several methods (lines 348-364). We comment on how this stoichiometric shift may contribute to intrinsic respiratory activation despite reduced mitochondrial content.

**Recommendations for the authors:**

**Reviewer #2 (Recommendations for the authors):**
Further Minor points.(9) It would be helpful to know further details regarding the source of the tumour samples, particularly for the proteomics (N=5) and transcriptomics (N=6) datasets, since the exact timepoint of tissue harvest and number of tumours/mouse varied, according to the methods section. Were all samples from the omics studies from different mice (ie 11 mice)? B4 and B6 seem like outliers in mitochondrial transcriptomes. Are these directly paired eg with T4 and T6? Are the side-by-side pairs of Ben and Tum samples for blots in Figure 1 and Supplementary Figure 1 from the same mouse.

This has been clarified in the Methods section (lines 424-429). Mitochondrial respiration experiments were performed in paired tissue (benign and tumor from same mouse). Transcriptomic, proteomic and histological analyses were performed on tumors and benign samples from different mice due to tissue limitations.

(10) Further references and details are needed to support the methodology of the mitochondrial function tests (eg. nutrients vs pairing with complexes). What was the time point of nutrient supplementation? It would seem that the lipid substrates should take longer to activate OXPHOS than pyruvate/malate or succinate. Is this the case? Is there speculation as to why succinate supplementation is much more active than pyruvate+malate? What is +MD in Figure 6? The rationale for pooling data for Figure 7A is unclear since the categories appear to overlap: (pyruvate, malate, ADP) vs. (palmitoyl-carnitine, malate, ADP).

Thank you for this comment. We have expanded the methods (lines 515-531) to provide additional detail on the mitochondrial respiration protocol. Briefly, permeabilized tissues were exposed to substrates delivered at supraphysiological concentrations in a sequential protocol lasting ~30–60 minutes. Under these conditions, mitochondrial respiration reflects the maximal capacity to utilize each substrate rather than the physiological time course of substrate mobilization or uptake that would occur in vivo with the influence of blood flow and transport/substrate availability limitations.

(11) Many of the figures were blurry (Figure 1F, 2B) or had labels that were too small to be effective (Figures 1G, H, 2D-G, 3E-G, 5E-I, 7C, 8B).

The font size of figure labels has been increased where possible and all figures have been exported to maximize resolution.